# DEEP INDEPENDENT VECTOR ANALYSIS

## ABSTRACT

We introduce a deep multivariate latent variable model, Deep Independent Vector Analysis (DeepIVA), for learning *linked* and *identifiable* latent sources across multiple data modalities by unifying multidataset independent subspace analysis (MISA) and identifiable variational autoencoders (iVAE). DeepIVA aims to learn hidden linkage information via the MISA loss to attain latent cross-modal alignment while leveraging the identifiability properties of the iVAE to ensure proper unimodal disentanglement. We propose a stricter set of performance measures, facilitating comprehensive evaluation. We demonstrate that DeepIVA can successfully recover nonlinearly mixed multimodal sources on multiple synthetic datasets compared with iVAE and MISA. We then apply DeepIVA on a large multimodal neuroimaging dataset, and show that DeepIVA can reveal linked imaging sources associated with phenotype measures.

## 1 INTRODUCTION

One fundamental problem in representation learning is how to learn the latent variables used to generate the data. In blind source separation (BSS) (Silva et al., 2016), independent component analysis (ICA) (Comon, 1994) aims to recover latent sources that are statistically independent, but there is no guarantee of identifiability in general without additional assumptions. Notably, the solution of a *linear* ICA problem is identifiable only when at most one of latent sources is Gaussian (Comon, 1994). The solution of a *nonlinear* ICA problem, on the other hand, is highly non-unique without additional restrictions (Hyvärinen & Pajunen, 1999). If the learned sources are not identifiable, it is impossible to reveal the underlying structure of the data. Recent advancements in nonlinear ICA theory have proposed to recover identifiable latent sources mixed nonlinearly up to trivial indeterminacies by introducing auxiliary information (Hyvarinen & Morioka, 2016; Hyvarinen et al., 2019; Khemakhem et al., 2020). Specifically, an identifiable variational autoencoder (iVAE) (Khemakhem et al., 2020) has been proved to recover nonlinearly mixed sources up to permutations or sign flips by utilizing auxiliary variables such as time indices or class labels. It assumes that sources are conditionally independent given such auxiliary variables, in the form of an exponential family distribution.

Apart from identifiability, we are often interested in learning linked representations from multiple data modalities, as each modality can only capture limited information of the data-generating system. For example, in the field of neuroimaging, structural magnetic resonance imaging (sMRI) can reveal static anatomical structure of the brain in high resolution, while functional magnetic resonance imaging (fMRI) can capture temporal dynamics at the cost of lower spatial resolution. Jointly analyzing two imaging modalities can uncover cross-modal relationships that cannot be detected by a single imaging modality, providing new insights into structural and functional interactions in the brain and its disorders (Calhoun & Sui, 2016). Recent studies on multi-view BSS assume that observations from different views originate from a shared source variable and distinct additive noise variables (Richard et al., 2020; 2021; Pandeva & Forré, 2023; Gresele et al., 2020). However, in the context of multimodal fusion, it is more reasonable to assume that each modality is generated by modality-specific latent variables which, in turn, are *linked* across modalities, rather than a *shared* set, especially for data modalities that are inherently heterogeneous.

To identify linked sources from multiple datasets, a unified framework called multidataset independent subspace analysis (MISA) has been developed (Silva et al., 2020) encompassing multiple *linear* latent variable models, such as ICA (Comon, 1994), independent vector analysis (IVA) (Kim et al., 2006), and independent subspace analysis (ISA) (Cardoso, 1998). MISA can be applied to analyze

both multi-subject and multimodal neuroimaging data. Built upon MISA, multimodal IVA (MMIVA) (Silva et al., 2021) and multimodal subspace IVA (MSIVA) (Li et al., 2023a) have been recently developed to capture one-to-one and many-to-many latent multimodal associations, respectively. In both cases, the learned linked latents are found to be significantly associated with phenotype measures such as age, sex and psychosis from large-scale multimodal neuroimaging datasets including sMRI and fMRI. Although both MMIVA and MSIVA assume that sources undergo a linear mixing process, it is possible that the true mixing process in neuroimaging data is actually *nonlinear*, considering nonlinear transformations in modeling and preprocessing stages. For example, the hemodynamic response function that models the relationship between neural activities and fMRI signals is nonlinear; preprocessing steps such as coregistration include nonlinear transformations. Nonlinear methods such as deep neural networks (LeCun et al., 2015) have been increasingly applied for neuroimaging data analysis, showing the potential to learn robust brain-phenotype relationships (Abrol et al., 2021).

Here, we ask the question: How can we learn *linked* and *identifiable* latent sources that are nonlinearly mixed across multiple data modalities? Built upon MISA and iVAE, we develop a deep multivariate latent variable model, Deep Independent Vector Analysis (DeepIVA), to learn linked and identifiable latent sources from multiple data modalities. In DeepIVA, we utilize the iVAE to identify sources from each modality, and the MISA loss function to align sources across all modalities. We demonstrate that DeepIVA can effectively recover sources compared to iVAE and MISA on multiple synthetic datasets and a large multimodal neuroimaging dataset. Our key contributions are as follows:

- We propose a deep latent variable model, DeepIVA, to learn linked and identifiable representations from multimodal data by unifying MISA and iVAE;
- We propose multiple evaluation metrics, including segment-specific minimum distance and trimmed mean correlation coefficient, to comprehensively characterize model performance;
- We perform a systematic evaluation of model performance and demonstrate that DeepIVA can effectively learn linked and identifiable multimodal sources in multiple simulation configurations (different sources, segments, and observations per segment);
- We apply DeepIVA on a large multimodal neuroimaging dataset to identify biologically meaningful sources associated with phenotype measures (age and sex).

## 2 METHODS

### 2.1 DEEP INDEPENDENT VECTOR ANALYSIS

**Independent Vector Analysis** Independent vector analysis (IVA) (Kim et al., 2006) is a multivariate latent variable model which extends the ICA problem from a single dataset to multiple datasets and captures statistical dependence across datasets. IVA aims to identify linked *vector* sources across $M$ datasets or data modalities ($M > 1$) where each observation $\mathbf{x}^m$ can be modeled as a linear mixture $\mathbf{A}^m$ of statistically independent sources $\mathbf{s}^m$:

$$\mathbf{x}^m = \mathbf{A}^m \mathbf{s}^m, \tag{1}$$

where $\mathbf{x}^m \in \mathbb{R}^V$ is an observation in the $m$-th dataset or data modality $\mathbf{X}^m \in \mathbb{R}^{N \times V}$, $\mathbf{s}^m \in \mathbb{R}^C$ is the source corresponding to the observation $\mathbf{x}^m$, $\mathbf{A}^m \in \mathbb{R}^{V \times C}$ is the invertible linear mixing matrix, $m \in [\![1, M]\!]$ indexes the dataset or data modality, $N$ is the number of observations, $V$ is the number of features, and $C$ is the number of sources ($C \leq V$). Particularly, in neuroimaging data, the observations are the subjects and the features are the volume pixel (voxel) intensities.

The IVA algorithm seeks to identify the sources $\hat{\mathbf{s}}^m$ by learning a demixing matrix $\mathbf{W}^m$: $\hat{\mathbf{s}}^m = \mathbf{W}^m \mathbf{x}^m$. The IVA problem can be solved by minimizing the following mutual information loss (Adali et al., 2014):

$$\mathcal{L}_{\text{IVA}} = \sum_{i=1}^{C} \left( \sum_{m=1}^{M} H(\mathbf{s}_i^m) - \mathcal{I}(\mathbf{s}_i) \right) - \sum_{m=1}^{M} \log |\det \mathbf{W}^m|, \tag{2}$$

where $H(\cdot)$ denotes the entropy, $\mathcal{I}(\cdot)$ denotes the mutual information, $\mathbf{s}_i$ is the $i$-th source component vector (SCV) which spans $M$ datasets, $\mathbf{s}_i = [s_i^1, s_i^2, \ldots, s_i^m]^\top$. The IVA objective aims to minimize the mutual information among SCVs while capturing multimodal dependence among sources within each SCV.

**Multidataset Independent Subspace Analysis** Multidataset independent subspace analysis (MISA) (Silva et al., 2020) is a unified framework encompassing multiple *linear* BSS models including ICA, IVA and ISA. MISA utilizes a multivariate Kotz distribution (Kotz, 1975) for SCV modeling:

$$p_\psi(\mathbf{s}_i) = \frac{\beta \lambda^\nu \Gamma\left(\frac{d_i}{2}\right) \left(\mathbf{s}_i^\top \mathbf{D}_i^{-1} \mathbf{s}_i\right)^{\eta-1}}{\pi^{\frac{d_i}{2}} \left(\det \mathbf{D}_i\right)^{\frac{1}{2}} \Gamma(\nu)} e^{-\lambda \left(\mathbf{s}_i^\top \mathbf{D}_i^{-1} \mathbf{s}_i\right)^\beta}, \tag{3}$$

where $\psi = [\beta, \lambda, \eta]$ is the set of Kotz hyperparameters, and $d_i$ is the $i$-th SCV dimension, here $d_i = M$. We define $\nu \triangleq \frac{2\eta + d_i - 2}{2\beta} > 0$ and $\alpha \triangleq \frac{\Gamma(\nu + \beta^{-1})}{\lambda^{\beta^{-1}} d_i \Gamma(\nu)}$ for brevity, where $\Gamma(\cdot)$ denotes the gamma function. The positive definite dispersion matrix $\mathbf{D}_i$ is related to the SCV covariance matrix $\mathbf{\Sigma}_{\mathbf{s}_i}$ as $\mathbf{D}_i = \alpha^{-1} \mathbf{\Sigma}_{\mathbf{s}_i}$. The Kotz distribution is highly flexible, as it encompasses the multivariate Gaussian distribution ($\psi = [1, \frac{1}{2}, 1]$) and the multivariate Laplace distribution ($\psi = [\frac{1}{2}, 1, 1]$).

The MISA loss (Silva et al., 2020) is defined as the KL divergence between the joint distribution across all SCVs $p_\psi(\mathbf{s})$ and the product of the Kotz distributions from each SCV $p_\psi(\mathbf{s}_i)$:

$$\mathcal{L}_{\text{MISA}}(\mathbf{W}) = D_{\text{KL}}(p_\psi(\mathbf{s}) \| \prod_{i=1}^{C} p_\psi(\mathbf{s}_i))$$

$$= -\sum_{m=1}^{M} J_{D_m} + \frac{1}{2} \sum_{i=1}^{C} J_{C_i} - f - \sum_{i=1}^{C} \frac{\mu - 1}{N} \sum_{n=1}^{N} J_{F_{in}} + \sum_{i=1}^{C} \frac{\lambda}{N} \sum_{n=1}^{N} J_{E_{in}}, \tag{4}$$

where $J_{D_m} = \sum_{i=1}^{C} \ln |\sigma_i^m|$ and $\{\sigma_i^m\}_{i=1}^{C}$ is the set of non-zero singular values of the demixing matrix $\mathbf{W}^m$, $J_{C_i} = \ln|\det \mathbf{D}_i|$, $J_{F_i} = \ln(\mathbf{s}_i^\top \mathbf{D}_i^{-1} \mathbf{s}_i)$, $J_{E_i} = \ln(\mathbf{s}_i^\top \mathbf{D}_i^{-1} \mathbf{s}_i)^\beta$, $f = \sum_{i=1}^{C} \left[ \ln \beta + \nu \ln \lambda + \ln \Gamma\left(\frac{d_i}{2}\right) - \frac{d_i}{2} \ln \pi - \ln \Gamma(\nu) \right]$.

**Identifiable Variational Autoencoder** The original MISA framework only includes *linear* BSS methods. In practice, we are also often interested in learning *nonlinear* mixtures, especially for high-dimensional data such as neuroimaging. Recently, an identifiable variational autoencoder (iVAE) (Khemakhem et al., 2020) has been proposed to recover latent sources that are nonlinearly mixed by conditioning latents on auxiliary variables. It has also been proved that iVAE can recover independent conditional latent variables while maximizing the likelihood of generating the data, thus bridging the gap between iVAE and nonlinear ICA (see Appendix F in Khemakhem et al. (2020) for more details).

Consider the following conditional unimodal generative model (Khemakhem et al., 2020):

$$\mathbf{x}^m = \mathbf{f}^m(\mathbf{s}^m) + \epsilon^m, \quad m = 1, \dots, M, \tag{5}$$

$$p_{\theta^m}(\mathbf{x}^m, \mathbf{s}^m | \mathbf{u}) = p_{\mathbf{f}^m}(\mathbf{x}^m | \mathbf{s}^m) p_{\mathbf{T}^m, \lambda^m}(\mathbf{s}^m | \mathbf{u}), \tag{6}$$

$$p_{\mathbf{f}^m}(\mathbf{x}^m | \mathbf{s}^m) = p_{\epsilon^m}(\mathbf{x}^m - \mathbf{f}^m(\mathbf{s}^m)), \tag{7}$$

$$p_{\mathbf{T}^m, \lambda^m}(\mathbf{s}^m | \mathbf{u}) = \prod_i^{C} \frac{Q_i^m(s_i^m)}{Z_i^m(\mathbf{u})} \exp\left[ \sum_{j=1}^{k} T_{i,j}^m(s_i^m) \lambda_{i,j}^m(\mathbf{u}) \right], \tag{8}$$

where $\mathbf{x}^m \in \mathbb{R}^V$ and $\mathbf{u} \in \mathbb{R}^S$ are observed random variables, $\mathbf{s}^m \in \mathbb{R}^C$ ($C \leq V$) is a latent variable, $\epsilon^m \in \mathbb{R}^V$ is an independent modality-specific noise variable with probability density function $p_{\epsilon^m}(\epsilon^m)$, $\theta^m = (\mathbf{f}^m, \mathbf{T}^m, \lambda^m)$ is a set of parameters of the conditional generative model, and $\mathbf{f}^m : \mathbb{R}^C \to \mathbb{R}^V$ is a nonlinear mixing function. We assume that the prior on the latent variables $p_{\theta^m}(\mathbf{s}^m | \mathbf{u})$ is conditionally independent, and each unimodal source $s_i^m$ follows a univariate exponential family distribution given the auxiliary variable $\mathbf{u}$, where $Q_i^m$ is the base measure, $Z_i^m(\mathbf{u})$ is the normalizing constant, $\mathbf{T}_i^m = (T_{i,1}^m, \dots, T_{i,k}^m)$ are the sufficient statistics, $\lambda_i^m(\mathbf{u}) = (\lambda_{i,1}^m(\mathbf{u}), \dots, \lambda_{i,k}^m(\mathbf{u}))$ are the parameters depending on $\mathbf{u}$, and $k$ is the dimension of each sufficient statistic.

Given a dataset $\mathcal{D} = \{(\mathbf{x}^{m(n)}, \mathbf{u}^{(n)})\}_{n=1}^{N}$ with $N$ observations sampled from the generative model defined by Equations 5 6 7 8, the iVAE aims to learn the parameters $(\theta^m, \phi^m)$ that maximize the data generation likelihood by maximizing the evidence lower bound (ELBO):

$$\mathcal{L}_{\text{iVAE}}^m(\theta^m, \phi^m) = \mathbb{E}_{q_\mathcal{D}} \left[ \mathbb{E}_{q_{\phi^m}(\mathbf{s}^m | \mathbf{x}^m, \mathbf{u})} [\log p_{\theta^m}(\mathbf{x}^m, \mathbf{s}^m | \mathbf{u}) - \log q_{\phi^m}(\mathbf{s}^m | \mathbf{x}^m, \mathbf{u})] \right], \tag{9}$$

where $q_\mathcal{D}$ is the empirical distribution of the dataset $\mathcal{D}$; $p_{\theta^m}(\mathbf{x}^m, \mathbf{s}^m | \mathbf{u})$ is the observed conditional joint distribution; $q_{\phi^m}(\mathbf{s}^m | \mathbf{x}^m, \mathbf{u})$ is the approximated posterior. The reparameterization trick

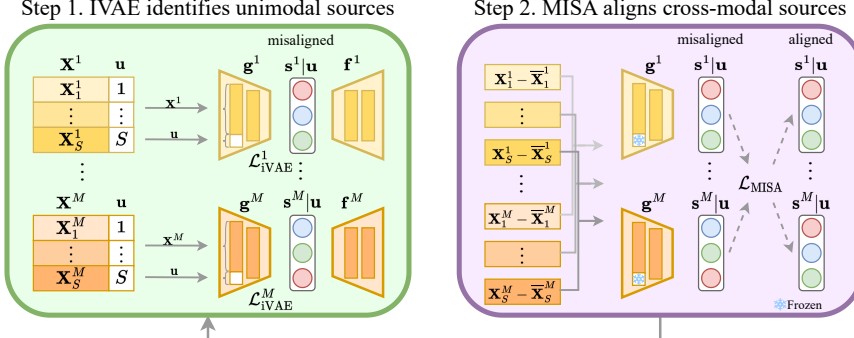

Step 1. IVAE identifies unimodal sources

Step 2. MISA aligns cross-modal sources

Iterate steps 1 and 2 until convergence

Figure 1: **DeepIVA overview.** Step 1: An iVAE is trained to recover sources for each of $M$ data modalities. Step 2: The MISA loss is applied to align sources across $M$ data modalities. Steps 1 and 2 are iterated until convergence.

(Kingma & Welling, 2013) is used to sample from a multivariate Gaussian distribution with a diagonal covariance, i.e. $q_{\phi^m}(\mathbf{s}^m|\mathbf{x}^m, \mathbf{u}) = \mathcal{N}\left(\mathbf{s}^m|\mathbf{g}^m(\mathbf{x}^m, \mathbf{u}; \phi_{\mathbf{g}^m}), \mathbf{I}\sigma^2(\mathbf{x}^m, \mathbf{u}; \phi_\sigma)\right)$.

We implement an $L$-layer multilayer perceptron (MLP) as the backbone of the iVAE. The input dimension of the first layer in the encoder is equal to the sum of the feature dimension and the auxiliary information dimension. The input and output dimensions of each intermediate layer are the same, which doubles the feature size ($2V$). The output dimension of the last layer is again equal to the feature dimension. We use Leaky ReLU (Andrew et al., 2013) as the activation function.

**Deep Independent Vector Analysis** Consider the following conditional multimodal generative model:

$$\mathbf{x}^m = \mathbf{f}^m(\mathbf{s}^m) + \epsilon^m, \quad m = 1, \dots, M, \tag{10}$$

$$p_\theta(\mathbf{x}^1, \dots, \mathbf{x}^M, \mathbf{s}^1, \dots, \mathbf{s}^M|\mathbf{u}) = \left(\prod_{m=1}^{M} p_{\mathbf{f}^m}(\mathbf{x}^m|\mathbf{s}^m)\right) p_{\theta_{\mathbf{s}}}(\mathbf{s}|\mathbf{u}), \tag{11}$$

where we define

$$p_{\mathbf{f}^m}(\mathbf{x}^m|\mathbf{s}^m) = p_{\epsilon^m}(\mathbf{x}^m - \mathbf{f}^m(\mathbf{s}^m)), \tag{12}$$

$$p_{\theta_{\mathbf{s}}}(\mathbf{s}|\mathbf{u}) = p_{\theta_{\mathbf{s}}}(\mathbf{s}^1, \dots, \mathbf{s}^M|\mathbf{u}) = \prod_{i=1}^{C} p_{\theta_{\mathbf{s},i}}(s_i^1, \dots, s_i^M|\mathbf{u}). \tag{13}$$

Integrating $p_{\theta_{\mathbf{s}}}(\mathbf{s}|\mathbf{u})$ over $s_i^{m'}$, $\forall i$, $\forall m', m' \neq m$, implies the following (marginal) conditionally independent unimodal latent model:

$$p_{\theta_{\mathbf{s}}^m}(s_1^m, \dots, s_C^m|\mathbf{u}) = \prod_{i=1}^{C} p_{\theta_{s,i}^m}(s_i^m|\mathbf{u}). \tag{14}$$

Built upon MISA and iVAE, we propose Deep Independent Vector Analysis (DeepIVA)[1] to learn *linked* and *identifiable* latent sources from multiple data modalities defined according to Equations 10 11 12 13 14 (Figure 1). Assuming the unimodal marginals $s_i^m|\mathbf{u}$ follow a univariate exponential family distribution, we show that the learned model parameters and sources from DeepIVA are identifiable up to a permutation and component-wise transformation (Appendix A).

In DeepIVA, an iVAE is first initiated for each data modality and then a single MISA module is initiated across all data modalities. The iVAE aims to recover sources for each modality and the MISA module aims to identify linkage of sources across modalities. At each epoch, we alternate between training the cross-modal MISA and the unimodal iVAEs. Specifically, we process one segment (segments are defined by the auxiliary variables) from all $M$ modalities at a time, and simultaneously update the encoder parameters for all modalities according to the MISA loss (Equation 4). We then

---

[1]Code will be made publicly available upon acceptance.

update the iVAE model parameters (both encoder and decoder) using all segments simultaneously, for each of the $M$ modalities separately, following the iVAE loss (Equation 9).

The MISA loss term $J_{D_m}$ in DeepIVA is different from the original MISA framework. Specifically, we compute the Jacobian matrix $\mathbf{J}^m$ of the nonlinear transformation parameterized by the MLP encoder $\mathbf{g}^m$ for the $m$-th data modality. For computational efficiency, we approximate the determinant of each Jacobian by the determinant of the average Jacobian across samples, $\overline{\mathbf{J}}^m = \frac{1}{N} \sum_{n=1}^{N} \frac{\partial \mathbf{g}^m}{\partial \mathbf{x}^{m(n)}}$. The loss term is defined as $J_{D_m} = \ln|\det \overline{\mathbf{J}}^m|$ if $\overline{\mathbf{J}}^m$ is a square matrix; $J_{D_m} = \sum_{i=1}^{C} \ln|\sigma_i^m|$ where $\{\sigma_i^m\}_{i=1}^{C}$ is the set of non-zero eigenvalues of $\overline{\mathbf{J}}^m \overline{\mathbf{J}}^{m\top}$ if $\overline{\mathbf{J}}^m$ is not a square matrix.

Additionally, since MISA is not designed to handle auxiliary information, we modify the original encoder architecture to distinguish between data features $\mathbf{x}^m$ and auxiliary variables $\mathbf{u}$ such that 1) the iVAE updates model parameters with respect to both $\mathbf{x}^m$ and $\mathbf{u}$ at the input layer, and 2) the MISA updates only those pertaining to $\mathbf{x}^m$ but not $\mathbf{u}$. The original iVAE model uses a single input layer taking the concatenated $\mathbf{x}^m$ and $\mathbf{u}$. In DeepIVA, we split this layer into two: one for data features $\mathbf{x}^m$ and another for auxiliary variables $\mathbf{u}$. The parameters with respect to $\mathbf{u}$ will only be updated at the iVAE training step but will remain frozen at the MISA training step. Also, the inputs for the auxiliary variables are set to 0 during MISA training to ensure no influence from the frozen weights.

## 2.2 SYNTHETIC DATA EXPERIMENT

**Synthetic Data** We generate multimodal synthetic datasets including non-stationary multivariate Gaussian sources. Specifically, we simulate a dataset $\mathbf{X} \in \mathbb{R}^{N \times C \times M}$ where $N = O \times S$ is the number of total observations, $O$ is the number of observations per segment, $S$ is the number of segments, $C$ is the number of sources, and $M$ is the number of modalities. Here, we set $M = 2$, $C \in \{5, 10, 15\}$, $S \in \{14, 8, 4\}$, $N \in \{2800, 5600\}$ to simulate real data, leading to 18 configurations in total. These configurations are chosen according to source identification performance in IVA tasks (Li et al., 2023b). For each segment, we generate a covariance matrix $\Sigma \in \mathbb{R}^{2C \times 2C}$ of both modalities, where the within-modality covariance matrices $\Sigma_{m,m} \in \mathbb{R}^{C \times C}$ along the main (block) diagonal are diagonal matrices with values sampled from a uniform distribution $[0.2, 4]$. Then, the between-modality covariance $\Sigma_{m,m'} \in \mathbb{R}^{C \times C}$ ($m \neq m'$) along the off-diagonal block is defined as a diagonal matrix with correlation values sampled from a uniform distribution $[0.7, 0.9]$ and scaled by the source standard deviations according to $\Sigma_{m,m}$. The data is then generated from a multivariate Gaussian distribution $\mathcal{N}(\mu, \Sigma)$, where $\mu \in \mathbb{R}^{2C}$ is sampled from a uniform distribution $[-3, 3]$. The auxiliary variable $\mathbf{u}$ is the segment label with a uniform distribution on the integer set $[\![1, S]\!]$. Latent variables within each modality are conditionally independent given segment labels $\mathbf{u}$. Synthetic sources are visualized in Appendix B.1. A neural network with $L = 2$ layers was employed to act as the nonlinear mixing function $h$. For each layer, a Leaky ReLU (Maas et al., 2013) with a negative slope of $0.2$ is used as the activation function. After the last Leaky ReLU layer, we multiply the mixed data from each modality by a different random orthogonal matrix $\mathbf{A}$ to obtain the final mixed dataset $\mathbf{X}$.

**Synthetic Data Experiment** For each configuration, we run iVAE, MISA and DeepIVA on the same synthetic data for 10 different random seeds, respectively. As for hyperparameters, we set an initial learning rate of $0.001$ for the iVAE model. The corresponding MISA learning rate is equal to the iVAE learning rate divided by the number of segments, considering that the MISA model is trained on data from each segment separately. A learning rate scheduler is used to reduce the learning rate by a factor of $0.1$ if there is no improvement for 20 epochs. We set the number of maximum contiguous iterations as 10 for both models. For synthetic datasets with 4, 8, and 14 segments, we use a batch size of 140, 160 and 160 for the iVAE model, and a batch size of 200, 350 and 700 for the MISA model, respectively. The model parameters are updated by the Adam optimizer (Kingma & Ba, 2014). Each model is trained for 300 epochs until convergence.

## 2.3 NEUROIMAGING DATA EXPERIMENT

**Neuroimaging Data** We utilize the UK Biobank dataset (Miller et al., 2016) $\mathbf{X} \in \mathbb{R}^{N \times V \times M}$ including two imaging modalities T1-weighted sMRI and resting-state fMRI ($M = 2$) from 2907 subjects ($N = 2907$). We preprocess sMRI and fMRI to obtain the gray matter tissue probability segmentation (GM) and amplitude of low frequency fluctuations (ALFF) feature maps, respectively. Each GM or ALFF feature map includes 44318 voxels ($V = 44318$). Here, we use age and sex groups

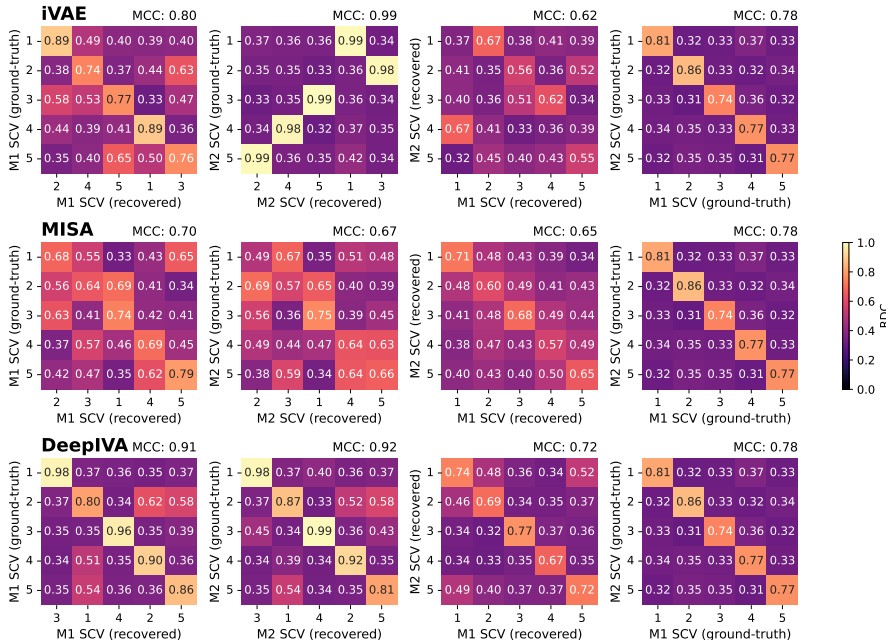

Figure 2: **Aggregated RDC matrices across segments from a synthetic dataset** (2800 **samples,** 5 **sources,** 14 **segments**). IVAE can correctly identify sources from each modality while MISA can better capture linked sources across both modalities. DeepIVA, which unifies iVAE and MISA, can not only recover unimodal sources, but also capture cross-modal linkage.

as auxiliary information, assuming that sources within each modality are conditionally independent given the age and sex group. This assumption is based on studies showing the significant impact of age and sex on both brain structure and function (Raz et al., 2004; Good et al., 2001; Ruigrok et al., 2014). We divide neuroimaging data into 14 segments according to sex and age groups such that segments approximately follow a uniform distribution (2 sex groups: male and female; 7 age groups: $46 - 53, 53 - 57, 57 - 61, 61 - 64, 64 - 67, 67 - 70, 70 - 79$ years old).

**Neuroimaging Data Experiment** We first run singular value decomposition on each data modality and choose the number of latent sources $C$ based on variance explained. We next apply multimodal group principal component analysis (MGPCA) on two data modalities (sMRI and fMRI) to reduce the feature dimension from $44318$ voxels to $C$ common sources. After that, the transformation is applied separately to each dataset in order to obtain modality-specific reductions. We next run iVAE, MISA and DeepIVA on the reduced data $\mathbf{X}_r \in \mathbb{R}^{N \times C \times M}$, respectively. During the training process, we use a full batch size of 2907 samples for both iVAE and MISA, an iVAE learning rate of $0.001$, a MISA learning rate of $7.14 \times 10^{-5}$, 300 epochs and 10 iterations per epoch.

## 2.4 EVALUATION METRICS

We utilize two metrics, the trimmed mean correlation coefficient between the 25th percentile and the 75th percentile (MCC) and the minimum distance (MD), to evaluate model performance. Unlike MCC, which only measures similarity along the main diagonal after permutation, MD also accounts for off-diagonal (dis)similarity. For each metric, we derive four types of coefficients: 1) a coefficient per modality, per segment; 2) an aggregated coefficient per modality; 3) an aggregated coefficient per segment; 4) a final aggregated coefficient across all modalities and segments.

We first compute the randomized dependence coefficient (RDC) matrix $\mathbf{R}$ (Lopez-Paz et al., 2013) between the recovered sources and the ground-truth sources for each modality and each segment. Note that we compute a RDC matrix for *each* segment separately, instead of computing it across all segments by convention. Our segment-specific RDC can more precisely characterize the data within each segment and effectively mitigate the noise introduced when all segments are taken

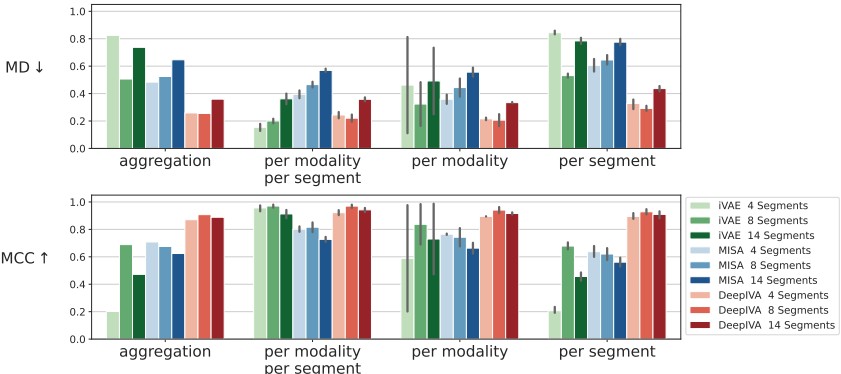

Figure 3: **MD (↓) and MCC (↑) values between recovered sources and ground-truth sources for three approaches (iVAE, MISA, DeepIVA) and three simulation configurations (2800 samples, 5 sources, 4/8/14 segments).** The bar plot shows the mean and 95% confidence interval. IVAE and DeepIVA show comparable performance for the per-modality per-segment metrics. DeepIVA demonstrates superior performance for the other metrics which account for unimodal identifiability, cross-modal linkage and cross-segment consistency.

simultaneously. Next, we aggregate the RDC matrices over segments by taking the mean to obtain an RDC matrix $\mathbf{R}^m$ *per modality* (mean aggregation). We also obtain an aggregated RDC matrix $\mathbf{R}^{[u]}$ *per segment* by taking the minimum across modalities for the entries corresponding to the sorted indices (i.e., the entries along the main diagonal after sorting) from a linear sum assignment problem (LSAP) solver (Crouse, 2016), and then taking the maximum for the remaining entries across all modalities (min-max aggregation). This min-max aggregation penalizes approaches that fail to detect cross-modal linkage, even when unimodal identifiability is high. To compute the final aggregated RDC matrix, we use min-max aggregation of $\mathbf{R}^m$ across modalities. We use the permuted indices from the modality-specific RDC matrix $\mathbf{R}^m$ which yields the lowest MD value as the *global* sorting indices to sort the other RDC matrices. For each sorted RDC matrix $\mathbf{R}_s$, we compute the MCC, as well as the MD, slightly adjusted from Equation 4 in Nordhausen et al. (2011):

$$\text{MD}(\mathbf{R}) = \frac{1}{2}(1 + \frac{1}{d}\text{trace}(\mathbf{R}\mathbf{R}^\top) - \frac{2}{d}\text{trace}(\mathbf{R}_s)), \tag{15}$$

where $\mathbf{R}$ is the unsorted matrix, $\mathbf{R}_s$ is the sorted matrix, and $d$ is the dimension of $\mathbf{R}$.

## 3 RESULTS

### 3.1 DEEPIVA LEARNS LINKED AND IDENTIFIABLE SOURCES FROM SYNTHETIC DATASETS

The aggregated RDC matrices for a synthetic dataset with 2800 samples, 5 sources and 14 segments from iVAE, MISA, and DeepIVA are shown in Figure 2. The aggregated RDC matrices for datasets with 4 and 8 segments are shown in the Appendix B.2 Figures 10 and 11. Columns I and II show the RDC matrices between the ground-truth sources and the recovered sources for the first modality (M1) and the second modality (M2), respectively. If an approach can successfully recover the latent sources that match the ground-truth sources, we anticipate that high RDC values align along the main diagonal after column permutation (same for both modalities). Greater contrast indicates better source identification performance. Column III shows the RDC matrices of the recovered sources between two modalities, while column IV shows the RDC matrices of the ground-truth sources between two modalities. If an approach can successfully identify the cross-modal linkage, high RDC values will be aligned along the main diagonal in column III, as the ground-truth linkage pattern in column IV.

According to Figure 2, we observe that iVAE can identify sources with high RDC values within each modality (M1 MCC: 0.80, M2 MCC: 0.99; row I, columns I and II) but fail to capture cross-modal linkage (MCC: 0.62; row I, column III). By constrast, MISA reveals stronger cross-modal dependence along the main diagonal, suggesting its ability to detect cross-modal linkage (MCC: 0.65; row II, column III). However, MISA cannot fully recover unique unimodal sources (M1 MCC: 0.70, M2

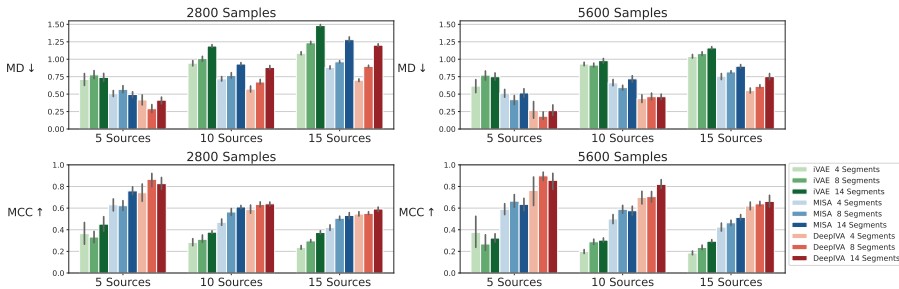

Figure 4: **Aggregated MD (↓) and MCC (↑) values between recovered sources and ground-truth sources in multiple synthetic data configurations.** The bar plot shows the mean and $95\%$ confidence interval across 10 random seeds. DeepIVA shows the best aggregated performance in all simulations.

MCC: 0.67; row II, columns I and II). In the first modality (M1), we note that the recovered SCV 1 shows high dependence with both ground-truth SCVs 2 and 3. DeepIVA, which unifies iVAE and MISA, can not only recover unimodal sources (M1 MCC: 0.91, M2 MCC: 0.92; row III, columns I and II) but also show the strongest cross-modal linkage (MCC: 0.72; row III, column III).

The corresponding MD and MCC measures are presented in Figure 3. The iVAE shows the best performance for the per-modality per-segment metrics (low MDs, high MCCs). As these metrics only account for identifiability within each modality and each segment (no aggregation), these results again indicate that iVAE can effectively recover segment-specific unimodal sources. We also note that DeepIVA achieves comparable performance to iVAE, suggesting that DeepIVA can also effectively identify sources. The other measures (coefficients per modality, coefficients per segment, and aggregated coefficients) take not only unimodal identifiability but also cross-segment consistency and cross-modal linkage into account. From these metrics, we observe that DeepIVA exhibits superior performance (lowest MDs, highest MCCs) over the other two approaches in all simulation configurations. The aggregated MDs from DeepIVA are consistently lower than those from iVAE and MISA across different segments. Specifically, for $4$, $8$, and $14$ segments, the aggregated MDs from DeepIVA are $68.62\%$, $49.59\%$, and $51.26\%$ lower than those from iVAE, respectively. Similarly, the aggregated MDs from DeepIVA are $46.49\%$, $51.37\%$, and $44.41\%$ lower than those from MISA for the corresponding segments. Furthermore, the aggregated MCCs from DeepIVA are consistently higher than those from iVAE and MISA. Notably, the aggregated MCCs from DeepIVA are $332.95\%$, $31.71\%$, and $88.25\%$ higher than those from iVAE for $4$, $8$, and $14$ segments, respectively. Likewise, the aggregated MCCs from DeepIVA are $22.93\%$, $34.29\%$, and $42.36\%$ higher than those from MISA for the respective segments. Additionally, when comparing performance across datasets with $4$, $8$ and $14$ segments, the configuration of $4$ segments and $700$ samples per segment shows the best source identification performance for the per-modality per-segment metrics. It suggests that variability in the dataset grows with the number of segments, making the optimization problem harder to solve.

We perform a systematic evaluation of model performance across different data-generating configurations by varying both the problem scale ($5$, $10$ and $15$ sources) and the sample size ($2800$ and $5600$ samples). The aggregated MD and MCC metrics are shown in Figure 4. Remarkably, DeepIVA outperforms iVAE and MISA in *every* configuration, showcasing its superior performance across all evaluated scenarios. Within each panel, we observe a consistent drop in model performance as the number of latent sources increases, suggesting that the optimization problem becomes more challenging as the latent dimension increases. Across horizontal panels, the DeepIVA performance improves for configurations with $10$ and $15$ sources when the sample size increases from $2800$ to $5600$, indicating that a larger sample size is necessary to better recover sources in a harder problem.

### 3.2 DEEPIVA RECOVERS LINKED NEUROIMAGING SOURCES ASSOCIATED WITH SEX AND AGE

We run iVAE, MISA and DeepIVA on a multimodal neuroimaging dataset to evaluate their effectiveness in real data. Results from singular value decomposition of sMRI GM and fMRI ALFF feature maps suggest that top $15$ sources can capture a large portion of variance explained in the data (Appendix C.1 Figure 12), and thus we choose to identify $15$ common independent sources. The aggregated RDC matrices across segments between two neuroimaging modalities are presented in

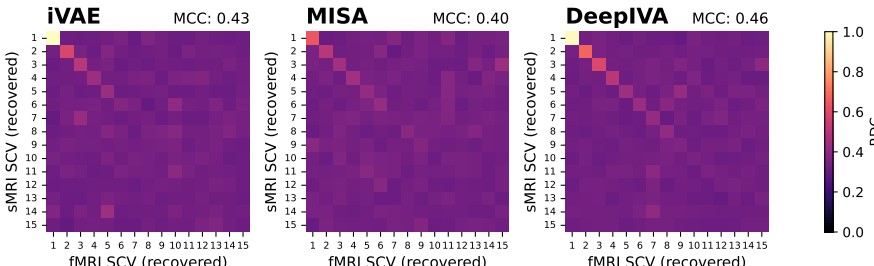

Figure 5: **Aggregated RDC matrices across** 14 **segments of** 15 **recovered sources between two imaging modalities.** DeepIVA captures cross-modal linkage from multimodal neuroimaging data.

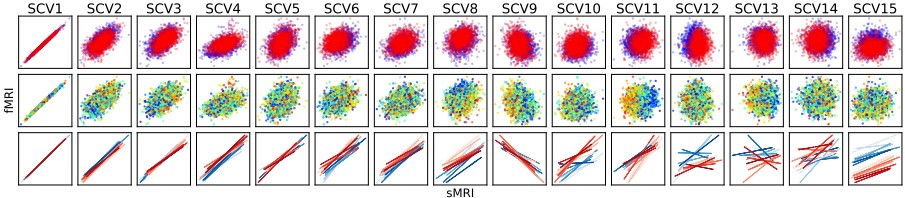

Figure 6: **DeepIVA linked imaging SCVs associated with sex and age.** Row I shows sex effect (blue: male; red: female). Row II shows aging effect (cold color: younger group; warm color: older group). Row III shows fitted linear lines from each segment (blue: male; red: female; light: younger group; dark: older group).

Figure 5. Similar to simulations, DeepIVA shows the strongest cross-modal dependence along the main diagonal (MCC: $0.46$), suggesting that it can better capture linked sources across two imaging modalities. We then color code the recovered sources from DeepIVA by sex and age groups (Figure 6), and observe noticeable sex clusters (e.g. SCVs $12$ and $15$) and age clusters (e.g. SCVs $8$ and $11$), indicating that DeepIVA captures linked sources related to phenotype measures. Furthermore, we fit a separate linear line for observations from each segment. If DeepIVA is capable of identifying consistent linked sources across segments, we should be able to observe that these fitted lines share similar slopes. Indeed, we note that slopes of fitted lines per segment are very consistent for most sources (e.g. SCVs $1 - 9$). Color-coded sources from iVAE are less aligned across segments while those from MISA are not associated with sex and age (Appendix C.2 Figures 13 and 14).

## 4 DISCUSSION

**Summary** We propose a deep multivariate latent variable model, Deep Independent Vector Analysis (DeepIVA), to learn linked and identifiable latent sources that are nonlinearly mixed across multiple data modalities. DeepIVA unifies iVAE and MISA, and exhibits unique advantages from each approach, specifically unimodal source identification from iVAE as well as cross-modal linkage detection from MISA. We demonstrate that DeepIVA can recover linked and identifiable sources from multiple synthetic datasets. Moreover, we show that DeepIVA reveals biologically meaningful linked sources from a large multimodal neuroimaging dataset.

**Limitations** DeepIVA assumes that sources are conditionally independent given the auxiliary variable to achieve identifiability, as it utilizes the iVAE objective. However, there may not be sufficient information about such an auxiliary variable in real data. Though we obtain sources related to age and sex groups, the true data-generating process remains unknown in the neuroimaging data.

**Future Work** We plan to extend our proposed method from nonlinear IVA problems to nonlinear ISA problems, aiming to capture source dependence by leveraging higher-dimensional subspaces. It is also worth exploring approaches that do not require side information, such as applying structural sparsity (Zheng et al., 2022), learning latent clusters (Willetts & Paige, 2021; Jiang et al., 2016) or using a Gaussian mixture prior and a deep ReLU/Leaky-ReLU network (Kivva et al., 2022).

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

## A  IDENTIFIABILITY

Here, we provide a conceptual sketch of the proof that the learned model parameters and sources from DeepIVA are identifiable up to a permutation of a component-wise transformation.

*Proof.* We consider the following conditionally independent multimodal generative model:

$$\mathbf{x}^m = \mathbf{f}^m(\mathbf{s}^m) + \epsilon^m, \quad m = 1, \dots, M, \tag{16}$$

$$p_\theta(\mathbf{x}^1, \dots, \mathbf{x}^M, \mathbf{s}^1, \dots, \mathbf{s}^M | \mathbf{u}) = \left( \prod_{m=1}^M p_{\mathbf{f}^m}(\mathbf{x}^m | \mathbf{s}^m) \right) p_{\theta_\mathbf{s}}(\mathbf{s} | \mathbf{u}), \tag{17}$$

where we define

$$p_{\mathbf{f}^m}(\mathbf{x}^m | \mathbf{s}^m) = p_{\epsilon^m}(\mathbf{x}^m - \mathbf{f}^m(\mathbf{s}^m)), \tag{18}$$

$$p_{\theta_\mathbf{s}}(\mathbf{s} | \mathbf{u}) = p_{\theta_\mathbf{s}}(\mathbf{s}^1, \dots, \mathbf{s}^M | \mathbf{u}) = \prod_{i=1}^C p_{\theta_{\mathbf{s},i}}(s_i^1, \dots, s_i^M | \mathbf{u}). \tag{19}$$

Integrating $p_{\theta_\mathbf{s}}(\mathbf{s} | \mathbf{u})$ over $s_i^{m'}$, $\forall i$, $\forall m', m' \neq m$, implies the following (marginal) conditionally independent unimodal latent model:

$$p_{\theta_\mathbf{s}^m}(s_1^m, \dots, s_C^m | \mathbf{u}) = \prod_{i=1}^C p_{\theta_{s,i}^m}(s_i^m | \mathbf{u}). \tag{20}$$

Assuming the unimodal marginals $s_i^m | \mathbf{u}$ can be accurately modeled with a univariate exponential family distribution, we can write the following conditionally independent unimodal generative model:

$$p_{\theta^m}(\mathbf{x}^m, \mathbf{s}^m | \mathbf{u}) = p_{\mathbf{f}^m}(\mathbf{x}^m | \mathbf{s}^m) p_{\theta_\mathbf{s}^m}(\mathbf{s}^m | \mathbf{u}), \tag{21}$$

$$p_{\theta_\mathbf{s}^m}(\mathbf{s}^m | \mathbf{u}) = p_{\mathbf{T}^m, \lambda^m}(\mathbf{s}^m | \mathbf{u}) = \prod_i^C \frac{Q_i^m(s_i^m)}{Z_i^m(\mathbf{u})} \exp\left[ \sum_{j=1}^k T_{i,j}^m(s_i^m) \lambda_{i,j}^m(\mathbf{u}) \right], \tag{22}$$

where $\mathbf{x}^m \in \mathbb{R}^V$ and $\mathbf{u} \in \mathbb{R}^S$ are two observed random variables, $\mathbf{s}^m \in \mathbb{R}^C$ ($C \leq V$) is a latent variable, $\epsilon^m \in \mathbb{R}^V$ is an independent modality-specific noise variable with probability density function $p_{\epsilon^m}(\epsilon^m)$, $\theta^m = (\mathbf{f}^m, \mathbf{T}^m, \lambda^m)$ is a set of parameters of the conditional generative model, $\mathbf{f}^m : \mathbb{R}^C \to \mathbb{R}^V$ is a nonlinear mixing function, $Q_i^m$ is the base measure, $Z_i^m(\mathbf{u})$ is the normalizing constant, $\mathbf{T}_i^m = (T_{i,1}^m, \dots, T_{i,k}^m)$ are the sufficient statistics, and $\lambda_i^m(\mathbf{u}) = (\lambda_{i,1}^m(\mathbf{u}), \dots, \lambda_{i,k}^m(\mathbf{u}))$ are the parameters depending on $\mathbf{u}$, and $k$ is the dimension of each sufficient statistic.

**Definition A.1** (Identifiability). *Let $\mathcal{P} = \{p_\theta : \theta \in \Theta\}$ be a statistical model with parameter space $\Theta$. We say $\mathcal{P}$ is identifiable up to $\sim$ if:*

$$\forall \theta, \hat{\theta} \in \Theta, \ p_\theta = p_{\hat{\theta}} \Rightarrow \theta \sim \hat{\theta}. \tag{23}$$

**Definition A.2** (Equivalence Relation). *Let $\sim$ be the equivalence relation on $\Theta$ defined as follows:*

$$(\mathbf{f}, \mathbf{T}, \lambda) \sim (\hat{\mathbf{f}}, \hat{\mathbf{T}}, \hat{\lambda}) \Leftrightarrow \exists A, \mathbf{c} \ | \ \mathbf{T}(\mathbf{f}^{-1}(\mathbf{x})) = A\hat{\mathbf{T}}\hat{\mathbf{f}}^{-1}(\mathbf{x})) + \mathbf{c}, \ \forall \mathbf{x} \in \mathcal{X}, \tag{24}$$

*where $A \in \mathbb{R}^{Ck \times Ck}$ is an invertible matrix and $\mathbf{c} \in \mathbb{R}^{Ck}$ is a vector.*

If the unimodal generative models defined by Equations 21 18 22 follows additional assumptions, the learned model parameters and sources are identifiable up to trivial indeterminacies, as proved in Khemakhem et al. (2020). We restate the key assumptions and Theorems in Khemakhem et al. (2020) as follows.

**Theorem 1.** *(Khemakhem et al., 2020) Assume that we observe data sampled from a generative model defined by Equations 21 18 22, with parameters $\theta^m = (\mathbf{f}^m, \mathbf{T}^m, \lambda^m)$. The parameters $\theta^m$ are $\sim_A$-identifiable if the following assumptions hold:*

*(i) The set $\{\mathbf{x}^m \in \mathcal{X} | \phi_{\epsilon^m}(\mathbf{x}^m) = 0\}$ has measure zero, where $\phi_{\epsilon^m}$ is the characteristic function of the density $p_{\epsilon^m}$ defined in Equation 18.*

*(ii) The mixing function $\mathbf{f}^m$ in Equation 18 is injective.*

*(iii) The sufficient statistics $T_{i,j}^m$ in Equation 22 are differentiable almost everywhere, and $(T_{i,j}^m)_{1 \leq j \leq k}$ are linearly independent on any subset of $\mathcal{X}$ of measure greater than zero.*

*(iv) There exist $Ck + 1$ distinct points $\mathbf{u}^0, \ldots, \mathbf{u}^{Ck}$ such that the matrix $L = \left[ \lambda^m(\mathbf{u}^1) - \lambda^m(\mathbf{u}^0), \ldots, \lambda^m(\mathbf{u}^{Ck}) - \lambda^m(\mathbf{u}^0) \right]$ of size $Ck \times Ck$ is invertible.*

Proof of Theorem 1 can be found at Khemakhem et al. (2020) Supplementary Material B.2.

Let $\hat{\theta}^m = (\hat{\mathbf{f}}^m, \hat{\mathbf{T}}^m, \hat{\lambda}^m)$ be the learned parameters from some algorithm that approximates the marginal distribution of the observations from the $m$-th dataset. This Theorem says necessarily $\hat{\theta}^m \sim_A \theta^m$, where $A$ is a linear transformation. The recovered sources $\hat{\mathbf{s}}^m = \hat{\mathbf{g}}^m(\mathbf{x}^m)$ are equal to the true sources $\mathbf{s}^m = \mathbf{g}^m(\mathbf{x}^m)$ up to a linear transformation $(A)$ of a component-wise nonlinear transformation (with respect to $\mathbf{T}^m$ and $\hat{\mathbf{T}}^m$). We can further assume sufficient conditions to obtain identifiability up to a permutation of a component-wise nonlinear transformation ($\sim_P$), as stated in the following Theorems 2 and 3.

**Theorem 2.** *(Khemakhem et al., 2020) Assume the hypotheses of Theorem 1 hold, and that $k \geq 2$. The parameters $\theta^m$ are $\sim_P$-identifiable if the following assumptions hold:*

*(i) The sufficient statistics $T_{i,j}^m$ in Equation 18 are twice differentiable.*

*(ii) The mixing function $\mathbf{f}^m$ has all second order cross derivatives.*

Proof of Theorem 2 can be found at Khemakhem et al. (2020) Supplementary Material B.3.

**Theorem 3.** *(Khemakhem et al., 2020) Assume the hypotheses of Theorem 1 hold, and that $k = 1$. The parameters $\theta^m$ are $\sim_P$-identifiable if the following assumptions hold:*

*(i) The sufficient statistics $T_{i,1}^m$ are not monotonic.*

*(ii) All partial derivatives of $\mathbf{f}^m$ are continuous.*

Proof of Theorem 3 can be found at Khemakhem et al. (2020) Supplementary Material B.4.

In DeepIVA, the assumptions of Theorems 1 2 3 suffice for identifiability per dataset or data modality, and lead to $\sim_P$-identifiable model parameters and sources for each dataset or data modality. Although these assumptions suffice for identifiability in unimodal iVAEs, they provide no guarantee of cross-modal alignment. In addition, following iVAE estimation, arbitrary component-wise nonlinearities remain, and thus we can assume they yield latents that follow a Kotz distribution marginal (see Section 2.1). Consequently, the MISA loss can be utilized to align sources across datasets or data modalities, for example by means of combinatorial optimization to search for modality-specific source permutation matrices. Sources are still identifiable after applying such permutations and component-wise nonlinear transformations, according to Theorems 1, 2, and 3. This concludes the identifiability proof for the proposed DeepIVA. $\qquad\square$

In practice, given the poor scalability of combinatorial search when the number of modalities $M$ and the number of latent sources $C$ grow, we pursue numerical optimization of the MISA loss to promote learning of a component-wise nonlinear function that yields Kotz marginals. This also has the added benefit of promoting source alignment in tandem with identification via unimodal iVAE training.

# B SYNTHETIC DATA EXPERIMENT

## B.1 SYNTHETIC DATA VISUALIZATION

Here, we visualize the ground-truth source pairs in synthetic data generated from 4 segments (Figure 7), 8 segments (Figure 8), and 14 segments (Figure 9).

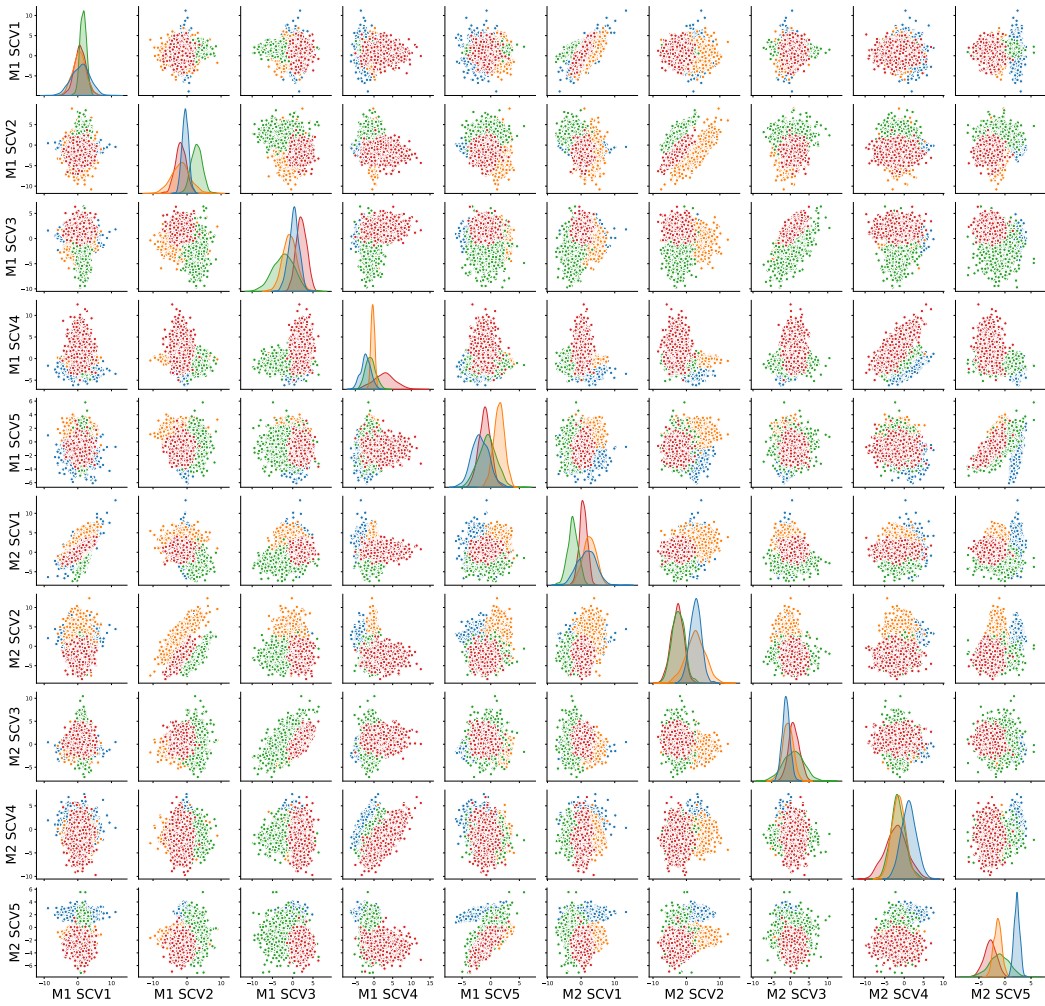

Figure 7: **Visualization of synthetic data (5 sources, 4 segments, 700 observations per segment).**

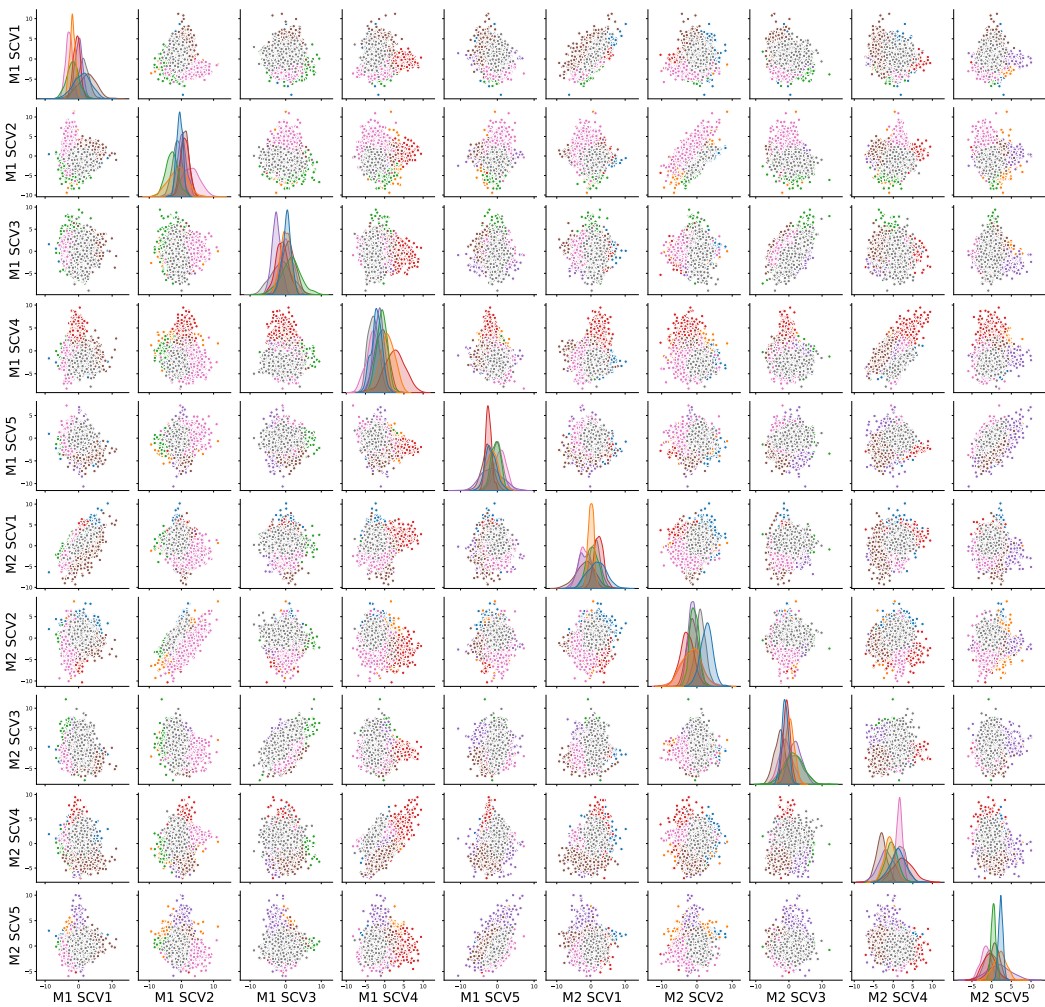

Figure 8: **Visualization of synthetic data (5 sources, 8 segments, 350 observations per segment).**

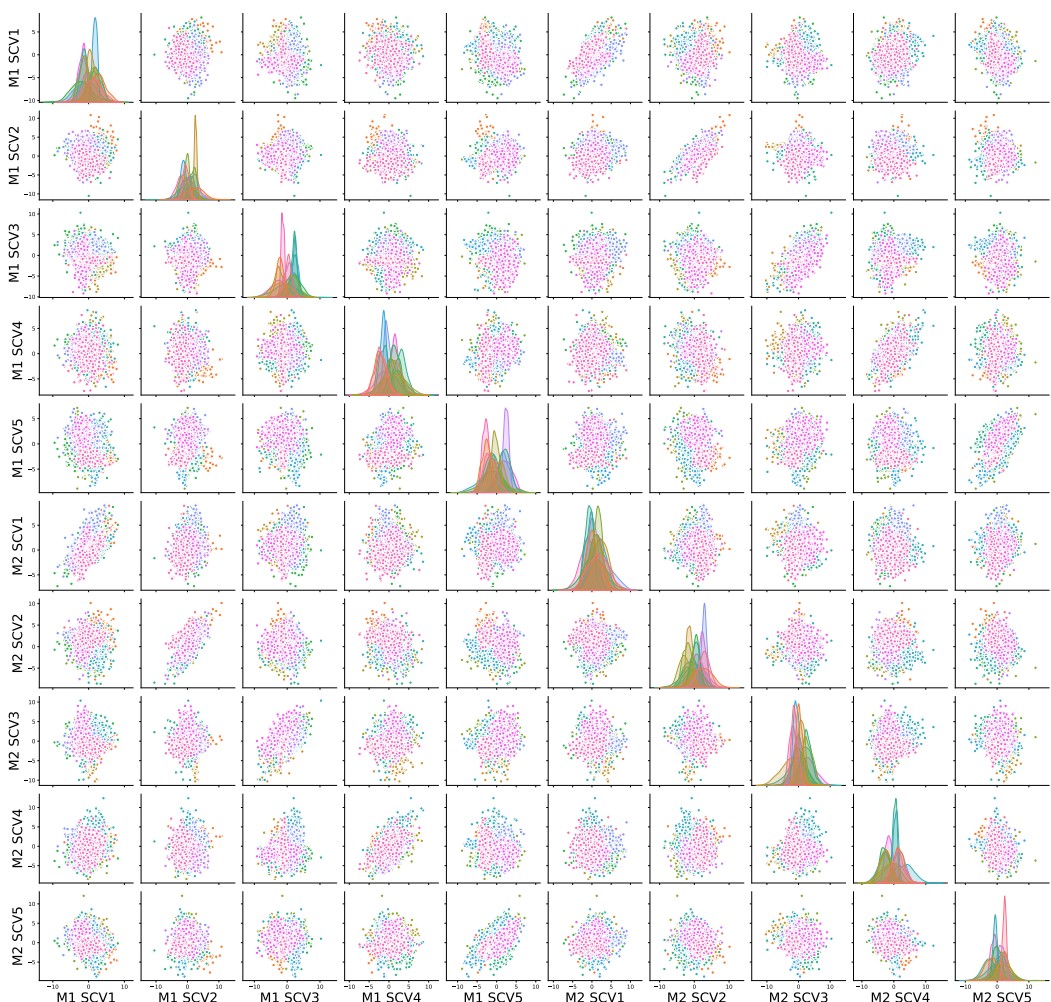

Figure 9: **Visualization of synthetic data (5 sources, 14 segments, 200 observations per segment).**

## B.2 Aggregated RDC Matrices

Here, we present the aggregated RDC matrices across segments for different configurations used to generate the data: 2800 samples, 5 sources and 4 segments (Figure 10); 2800 samples, 5 sources and 8 segments (Figure 11).

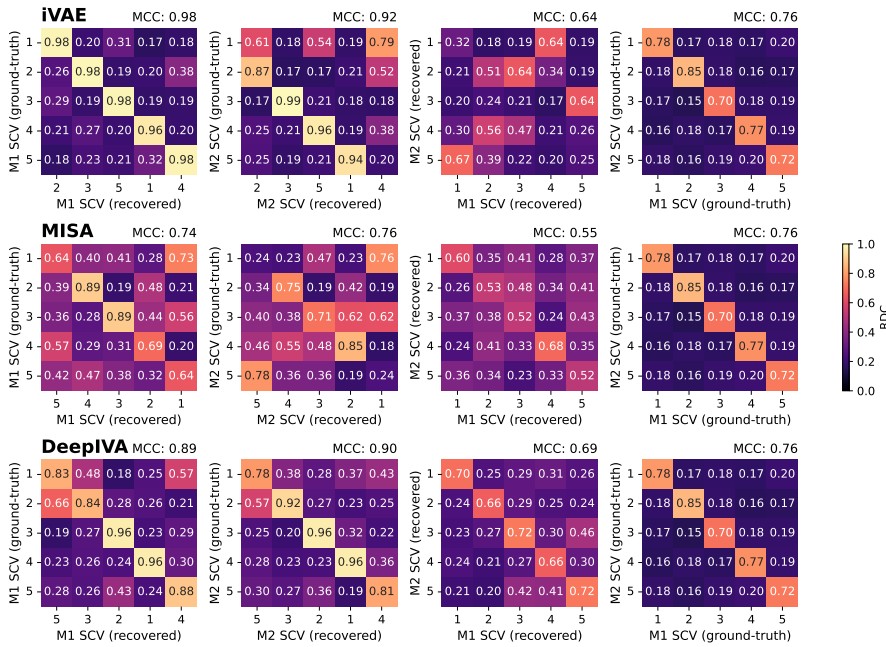

Figure 10: **Aggregated RDC matrices across segments from a synthetic dataset** (2800 **samples,** 5 **sources,** 4 **segments).**

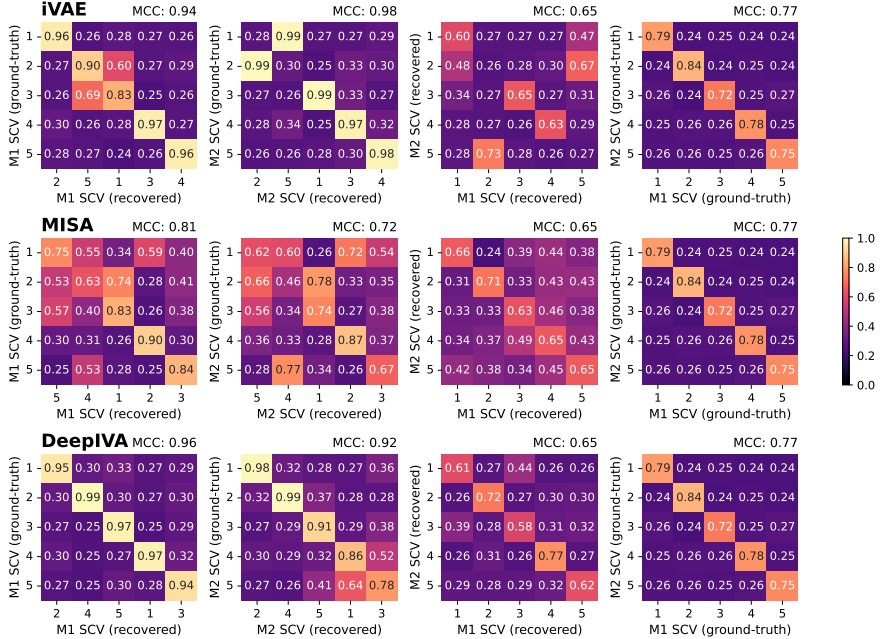

Figure 11: **Aggregated RDC matrices across segments from a synthetic dataset** (2800 **samples,** 5 **sources,** 8 **segments).**

## C NEUROIMAGING DATA EXPERIMENT

### C.1 SINGULAR VALUE DECOMPOSITION

We perform singular value decomposition on sMRI GM and fMRI ALFF feature maps, respectively. Figure 12 shows proportion of variance explained. According to the elbow criterion, we observe that $5 - 15$ sources can capture a large portion of variance explained. Thus, we choose $15$ latent sources for neuroimaging data.

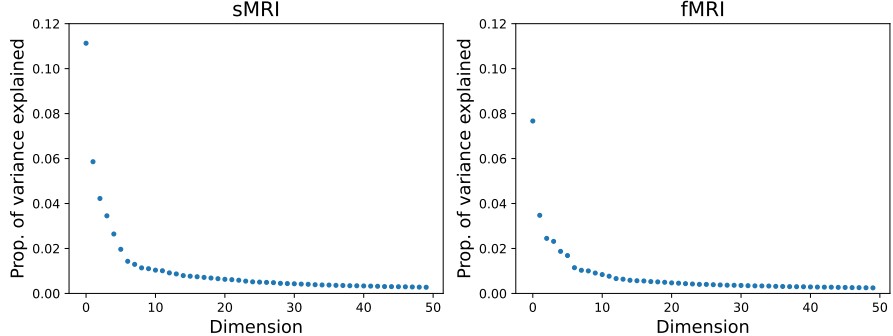

Figure 12: **Proportion of variance explained in neuroimaging data in top** $50$ **dimensions.**

Table 1: **Proportion of variance explained.**

| Dimension | sMRI | fMRI |
|:---------:|:-----:|:-----:|
| 5 | 0.273 | 0.178 |
| 10 | 0.342 | 0.236 |
| 15 | 0.389 | 0.270 |

### C.2 IMAGING SOURCES

We color code linked SCVs from neuroimaging data by sex and age, and present the color-coded SCVs from iVAE in Figure 13 and those from MISA in Figure 14. Row I shows sex effect (blue: male; red: female). Row II shows aging effect (cold color: younger group; warm color: older group). Row III shows fitted linear lines from each segment (blue: male; red: female; light: younger group; dark: older group).

In general, we observe that the iVAE SCVs reveal several sex clusters (e.g. SCVs 4 and 15) and age clusters (e.g. SCVs 5 and 15), but the linear fitted lines are less aligned across segments. By contrast, there is no visually obvious cluster related to sex and age from the MISA linked SCVs while the linear fitted lines are more aligned across segments. These results can be explained by the differences in the objectives of iVAE and MISA. Specifically, iVAE incorporates sex and age information during optimization, while MISA lacks access to such information. The loss function in MISA is specifically designed to align multimodal data, whereas iVAE is not explicitly designed for this purpose.

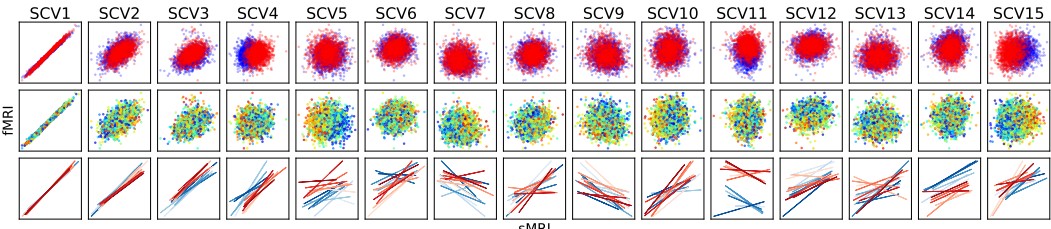

Figure 13: **IVAE imaging SCVs associated with sex and age.**

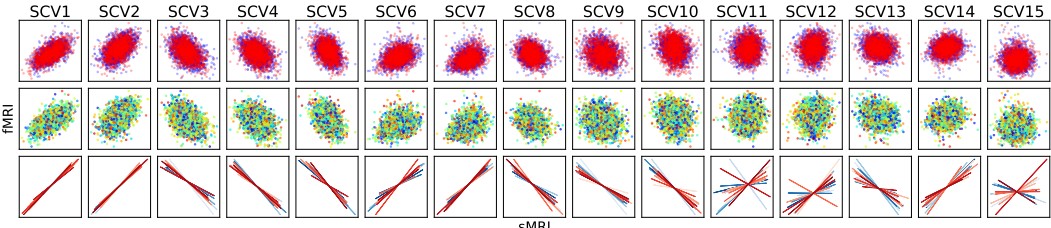

Figure 14: **MISA linked imaging SCVs associated with sex and age.**

