# A SYNTHETIC DATA EXPERIMENT

## A.1 SYNTHETIC DATA VISUALIZATION

Here, we visualize the ground-truth source pairs in synthetic data generated from 4 segments (Figure 8), 8 segments (Figure 9), and 14 segments (Figure 10).

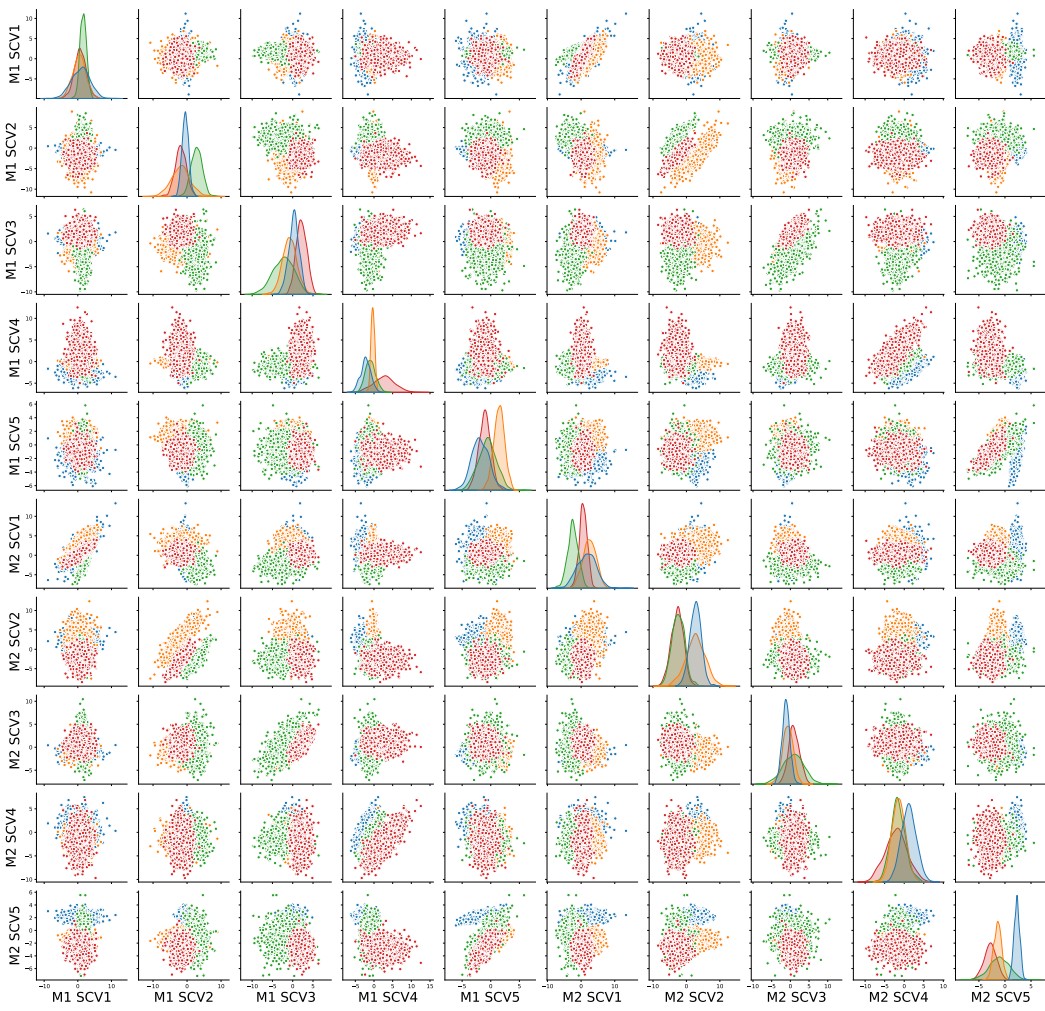

Figure 8: **Visualization of synthetic data (5 sources, 4 segments, 700 observations per segment).**

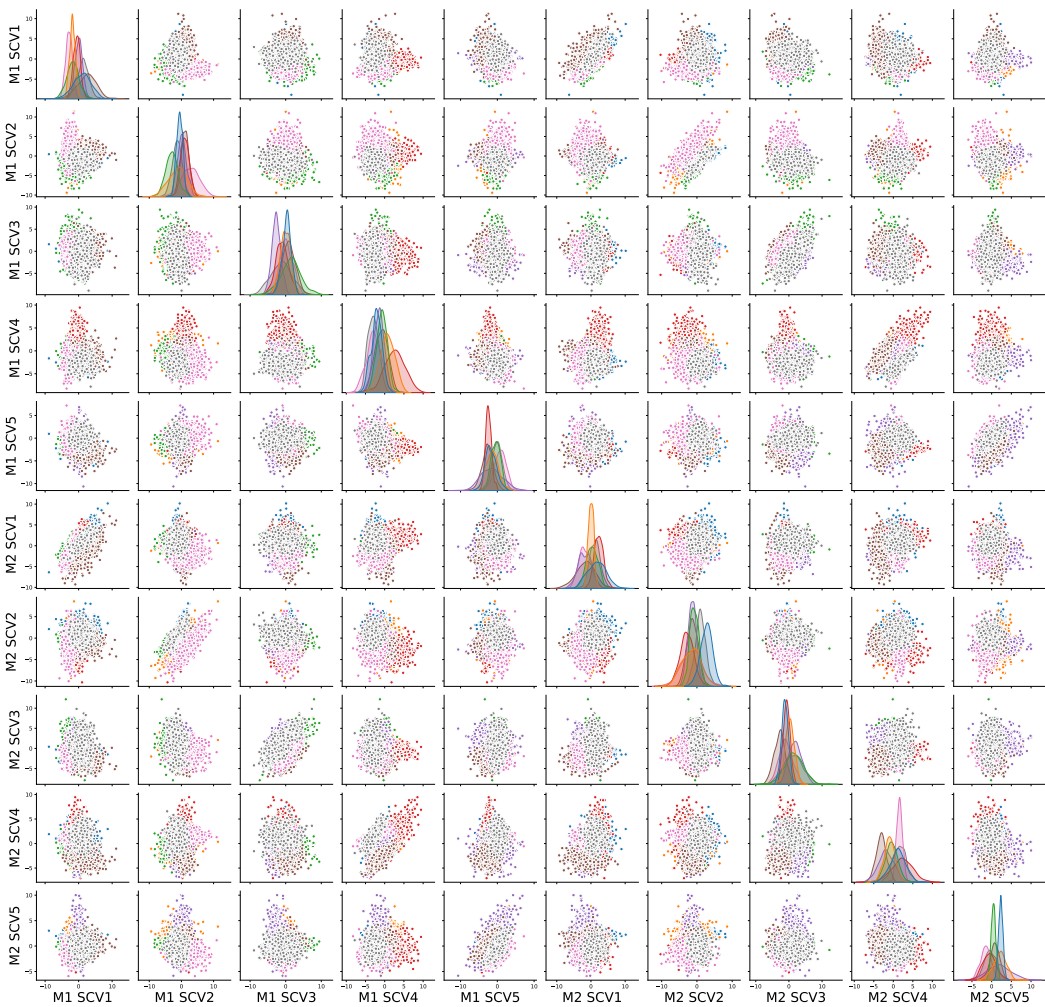

Figure 9: **Visualization of synthetic data (5 sources, 8 segments, 350 observations per segment).**

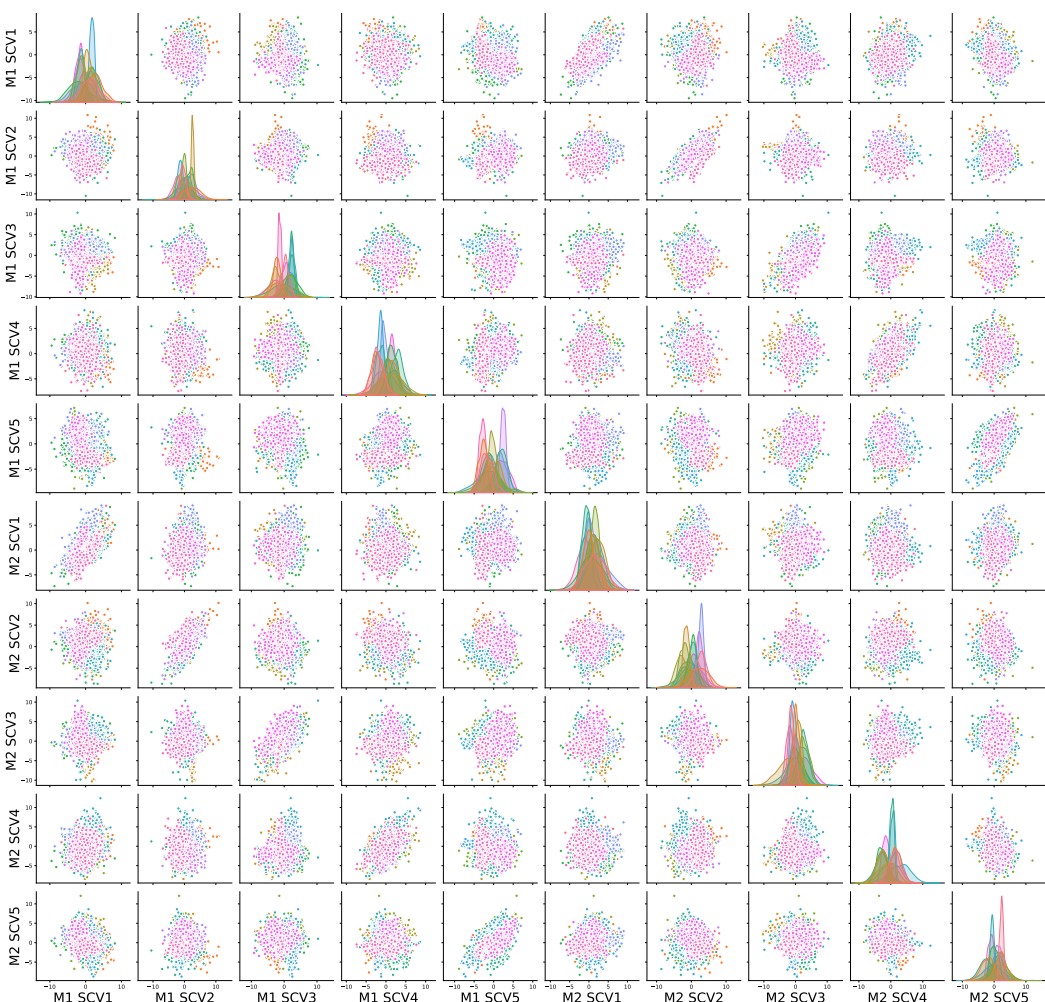

Figure 10: **Visualization of synthetic data (5 sources, 14 segments, 200 observations per segment).**

## A.2 Aggregated RDC Matrices

Here, we present the aggregated RDC matrices across segments for four different data configurations: 5 sources and 4 segments (Figure 11); 5 sources and 8 segments (Figure 12); 15 sources and 4 segments (Figure 13); 15 sources and 8 segments (Figure 14).

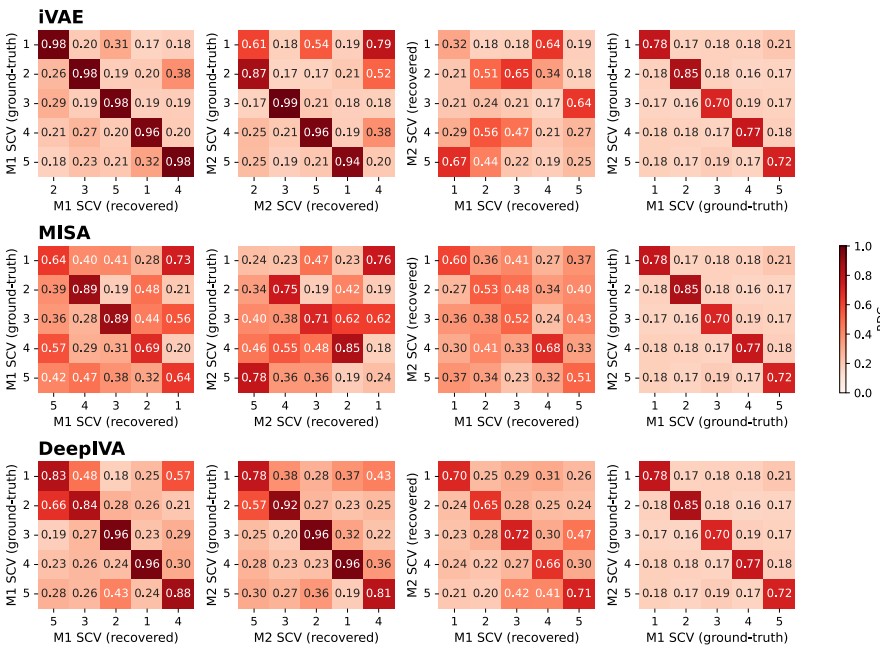

Figure 11: **Synthetic data (5 sources, 4 segments): Aggregated RDC matrices across segments between recovered sources and ground-truth sources.**

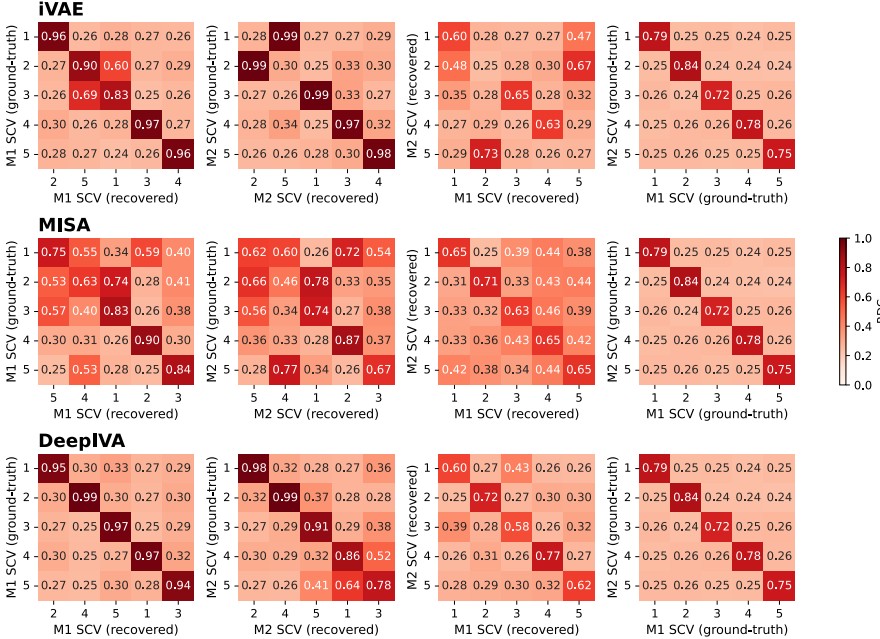

Figure 12: **Synthetic data (5 sources, 8 segments): Aggregated RDC matrices across segments between recovered sources and ground-truth sources.**

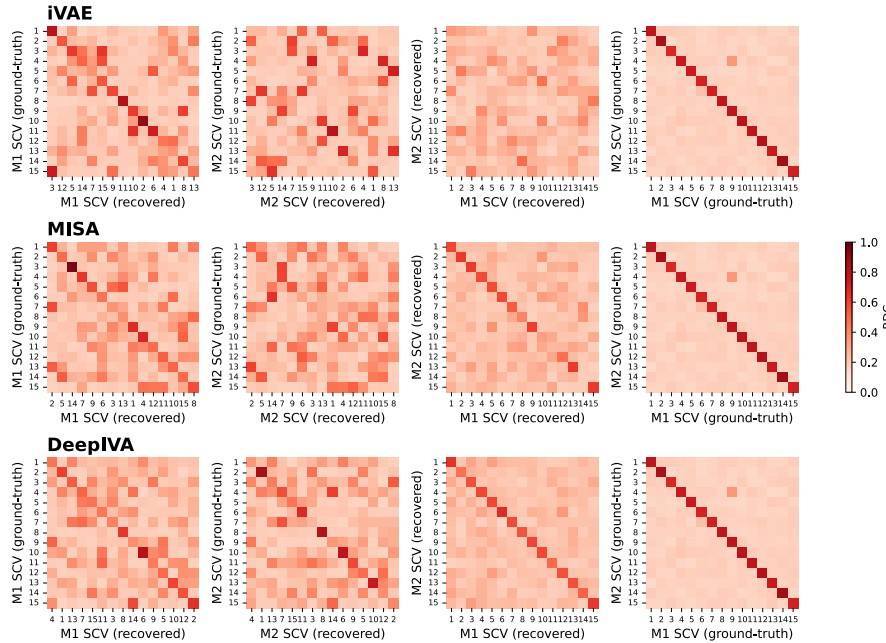

Figure 13: **Synthetic data** (15 **sources,** 4 **segments): Aggregated RDC matrices across segments between recovered sources and ground-truth sources.**

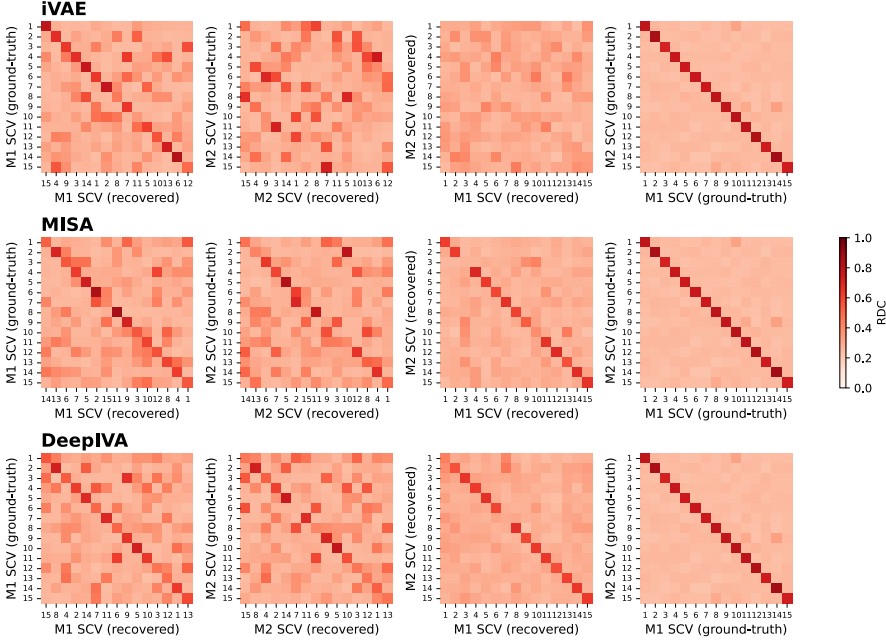

Figure 14: **Synthetic data** (15 **sources,** 8 **segments): Aggregated RDC matrices across segments between recovered sources and ground-truth sources.**

# B  NEUROIMAGING DATA EXPERIMENT

## B.1  SINGULAR VALUE DECOMPOSITION

We perform singular value decomposition on sMRI GM and fMRI ALFF feature maps, respectively. According to Figure 15, we observe that $10 - 20$ sources can capture most variance explained. Thus, we choose 15 latent sources for neuroimaging data.

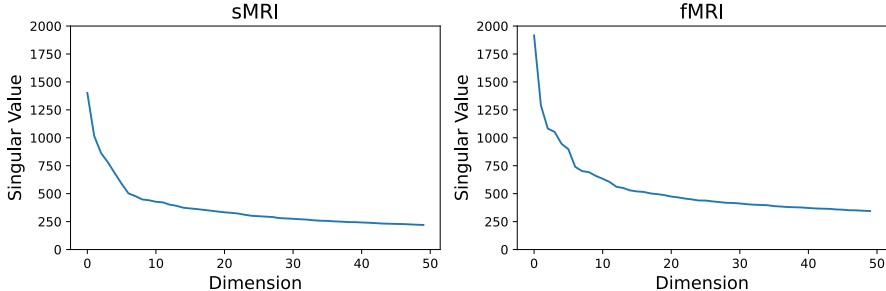

Figure 15: **Singular values of neuroimaging data.**