# OpenReview forum: "Deep Independent Vector Analysis"
_ICLR.cc/2024/Conference — Submitted to ICLR 2024_

### Official Review · Reviewer_FDhN · 2023-10-27

**Soundness:** 2 fair
**Presentation:** 3 good
**Contribution:** 2 fair
**Rating:** 5
**Confidence:** 2

**Summary:**

Authors propose DeepIVA, an approach for identifying non-linearly mixed sources across different datasets (modalities). DeepIVA combined the iVAE approach with the MISA algorithm, by utilizing a two-step training procedure. Authors propose a set of extended metrics to evaluate their proposed model against the iVAE and MISA baselines, and show that their approach achieves satisfying results in both the uni- and cross-modal settings, whereas the baselines perform well only in either of these. Furthermore experiments on fMRI data from UK Biobank are performed as an example of a real-life application.

**Strengths:**

- The proposed method performs well both in terms of cross-modal source identification, as well as in the unimodal setting - although for a higher number of segments iVAE is still better in the unimodal one
- Authors developed novel metric formulations for the cross modal setup, and described them in a clear manner
- The experimental setup is clearly described in the main text, along with all hyperparameter values

**Weaknesses:**

I am missing a more theoretically grounded justification for why the approach of combining MISA and iVAE would yield an identifiable solution in the cross-modal setting (especially with the alternating two-step training procedure, see Question 2). So far the approach seems like stitching existing methods together without further introspection. Pointing out to at least a sketch of a proof, e.g., in the appendix would be desirable.

**Questions:**

- While interesting, it is hard to interpret the soundness of the MRI results. UK Biobank contains a wide range of brain MRI-derived variables, e.g., volumetrical informations of different brain regions of interest (ROIs). Did the authors consider incorporating these into the analysis - perhaps as a proxy for ground-truth sources, since the underlying ROI volumes stay the same regardless of MRI type?
- Maybe I am missing something, but why does DeepIVA have to be trained with a two-step procedure? Since iVAE is identifiable, can’t we first train each separate modality-specific VAE and then align the sources using MISA?

---

> ### Author Response · Authors · 2023-11-20
>
> 1. Thank you for the suggestion on the sketch of proof. We have restated key assumptions and Theorems in the identifiability theory (Khemakhem et al. 2020), and provided a conceptual sketch of proof showing that the generative model assumed in DeepIVA is identifiable up to a permutation and component-wise transformation in Appendix A. We have also updated Section 2.1 iVAE and DeepIVA parts accordingly in the revised manuscript.
>
> 2. We are not clear about what the reviewer is suggesting we could do with the ROIs specifically. We aim to identify latent sources in a data-driven manner, and it is not necessary that linked sources are located at similar brain regions. Thus, a predefined ROI atlas might not be ideal as a proxy for ground-truth sources. Also, ROIs are not always the same for all types of MRI (for example, white matter tracts vs gray matter ROIs). While not seemingly a good depiction of the ground-truth, ROI-based data may be a good alternative to dimensionality reduction. We will consider that in future experiments.
>
> 3. Thank you for the thoughtful question. We train DeepIVA in a two-step procedure because of computational efficiency and practical consideration. If we train iVAE and then use MISA to align unimodal sources, it will require solving a combinatorial problem, which can be computationally expensive as the number of sources and datasets scale up. Additionally, if we first train iVAE independently, it is possible that parameters of one iVAE model might be stuck in local minima and the learned sources might not be linked. For example, in Figure 2 first row, we trained two iVAE models independently, and each model can identify sources fairly well, but cross-modal alignment is poor between two sets of independently identified sources. Iteratively training iVAE and MISA can guide the model to a solution not only maximizing for identifiability but also simultaneously maximizing for linkage.

---

> > ### Comment · Reviewer_FDhN · 2023-11-22
> >
> > Thank you for your response. I will retain my score, as the method is rather incremental at this point.

---

### Official Review · Reviewer_queK · 2023-10-30

**Soundness:** 3 good
**Presentation:** 3 good
**Contribution:** 2 fair
**Rating:** 5
**Confidence:** 3

**Summary:**

The paper suggests a combination of identifiable variational autoencoders (iVAEs) and multidataset independent subspace analysis (MISA). The approach iteratively (i) maximizes the iVAE EVBO separately for each modality; and (ii) minimizes the KL divergence between the joint conditional prior distribution of the latents and their marginal product. Experiments on synthetic data and neuroimaging data illustrate that the approach improves the latent identifiability and cross-modal linkages, compared to iVAE and MISA.

**Strengths:**

The idea of combining iVAE and MISA is new as far as I am aware. Using MISA to align the latent representation seems quite interesting and could be a promising alternative for example to self-supervised learning approaches such as [1,2]. The approach might be in particular useful (compared e.g. with contrastive approaches) for M>2 modalities. However, this has not been explored in the current experiments.

Numerical experiments on simulated data and a neuroimaging data show improvements for the inferred representations according to the considered evaluation measures, when compared with iVAE and MISA. The paper is largely well written and easy to follow.

[1] Lyu, Qi, et al. "Understanding Latent Correlation-Based Multiview Learning and Self-Supervision: An Identifiability Perspective." International Conference on Learning Representations. 2021.
[2] Daunhawer, Imant, et al. "Identifiability results for multimodal contrastive learning." arXiv preprint arXiv:2303.09166 (2023).

**Weaknesses:**

The submission claims to learn identifiable representation. However, I could not find a proof of it. It has also not been defined what this identifiability means in the multi-modal context. In particular, what are the equivalence classes? What are the necessary conditions required for identifiability? Does the availability of multiple datasets/modalities lead to less restrictive conditions? Do the conditions hold in practice?

Likewise, the submission claims to learn disentangled representations. I could not find a proof of this either. Is it obvious that iterating between training steps 1 and 2 yields a disentangled representation at convergence?

It is not clear to me how scalable the method is as it requires computing the log-determinant of the encoder Jacobian.

It is not clear to me how the MISA steps affects other generative performances that look not just at the latent variables of the model. For example, does it lead to worse LLH (or lower bounds thereof), FID score etc. compared to iVAE?

**Questions:**

To clarify, the iVAE models in the experiments mean training individual iVAEs for each modality and then learning a rotation matrix to align the latents (as done for the MCC evaluation under weak identifiability)?

How does the approach compare to a single iVAE model on the modality-concatenated data? This would learn shared/perfectly aligned sources.

Can you clarify how you have modified the encoder architecture to make sure for example that ‘MISA updates only the model weights pertaining to the input features but not the auxiliary variables’? Does this exclude any interaction between input features and auxiliary variables?

In 2.2, is it possible to generalise the seemingly very restrictive assumption that the data is independent within each modality?

Can you clarify what is the advantage of the introduced MC measure compared to the MCC measure that is often also based on RDC?

Do MISA and iVAE latent representations also lead to similar clusters of age or sex phenotpyes? Does the multimodal dimensionality reduction in the pre-processing step impact the identifiability?

---

> ### Author Response · Authors · 2023-11-20
> **Official Comment by Authors (1/2)**
>
> We appreciate that the reviewer acknowledged the strengths of our work and include our point-to-point response as follows.
>
> 1. Identifiable representation. We have restated key assumptions and Theorems in the identifiability theory (Khemakhem et al. 2020), and provided a conceptual sketch of the proof showing that the generative model assumed in DeepIVA is identifiable up to a permutation and component-wise transformation in Appendix A. We have also updated Section 2.1 iVAE and DeepIVA parts accordingly in the revised manuscript.
>
> 2. Disentangled representation. We have updated “disentangled representations” to “latent sources”.
>
> 3. Scalability. Note that we utilize an approximate Jacobian by taking the average across all samples for subsequent computation (i.e. eigendecomposition), instead of using the Jacobian of each sample. Such an approximation leads to similar results while also providing computational efficiency gains. Moreover, we can compute the Jacobian for all modalities in parallel. Lastly, the identifiability theory of iVAE extends trivially to flow models (as discussed in Khemakhem et al. 2020), which have simple Jacobian form and could enable better scalability.
>
> 4. Generative performance. The iVAE loss from DeepIVA is typically not lower (better) than the vanilla iVAE loss after training for the same number of epochs, but this issue can be potentially mitigated by training the iVAEs alone following alignment at the last step of DeepIVA. Once aligned, it is likely that iVAE training would stay at the same local basin and keep the source permutation unchanged. Yet another solution is to update only the decoder weights for iVAE training while training the MISA loss on the encoder weights. This would help compensate for any negative effects of MISA on reconstruction.
>
> 5. Yes, the reviewer’s understanding is mostly correct, except that we did not apply a rotation matrix on the iVAE recovered sources (the synthetic dataset meets the conditions for P-identifiability within each modality) when we plotted the RDC matrices, such as Figure 2. Instead, we merely applied the same permutation matrix to all modalities such that high MCC values were along the main diagonal when we computed per-segment per-modality MCC.
>
> 6. Thank you for the thoughtful question. Let each dataset be a $N \times V$ matrix where $N$ is the number of samples and $V$ is the number of features. There are two ways to concatenate data: One is to concatenate data along the sample dimension (as in joint ICA, learning a shared transformation $f$ for both modalities and different latents for each modality) and the other is to concatenate data along the feature dimension (as in group ICA, learning shared latents and a single transformation $f$ that mixes all modalities indistinctly from one another). Either way, we learn a shared nonlinear transformation across datasets. If datasets were generated from a shared latent variable, it might be reasonable to use one of these two approaches. However, we are interested in studying different heterogeneous data modalities generated from modality-specific, non-shared conditionally independent latent variables and modality-specific nonlinear mixing processes. Thus, it makes more sense to estimate sources for each dataset separately, as in IVA, utilizing iVAE to obtain identifiability and  MISA to capture linkage.
>
> 7. Sorry for the lack of clarification. Yes, our modification excludes the interaction between input features and auxiliary variables. We have updated Figure 1 and revised the related text in Section 2.1: “Additionally, since MISA is not designed to handle auxiliary information, we modify the original encoder architecture to distinguish between data features $\mathbf{x}^{m}$ and auxiliary variables $\mathbf{u}$ such that 1) the iVAE updates model parameters with respect to both $\mathbf{x}^{m}$ and $\mathbf{u}$ at the input layer, and 2) the MISA updates only those pertaining to $\mathbf{x}^{m}$ but not $\mathbf{u}$. The original iVAE model uses a single input layer taking the concatenated $\mathbf{x}^{m}$ and $\mathbf{u}$. In DeepIVA, we split this layer into two: one for data features $\mathbf{x}^{m}$ and another for auxiliary variables $\mathbf{u}$. The parameters with respect to $\mathbf{u}$ will only be updated at the iVAE training step but will remain frozen at the MISA training step. Also, the inputs for the auxiliary variables are set to $0$ during MISA training to ensure no influence from the frozen weights.”

---

> > ### Author Response · Authors · 2023-11-20
> > **Official Comment by Authors (2/2)**
> >
> > 8. Thank you for the interest in source dependence within each modality. In the future, we plan to generalize the independence assumption and utilize a higher-dimensional subspace structure to capture source dependence. At such time, the conditions for subspace identifiability will also be investigated accordingly. We have added discussion in Section 4 Future Work: “​​We plan to extend our proposed method from nonlinear IVA problems to nonlinear ISA problems, aiming to capture source dependence by leveraging higher-dimensional subspaces.”
> >
> > 9. Do you mean the minimum distance (MD) instead of MC? If so, the key benefit of MD is that it accounts for off-diagonal values (as in $\mathrm{trace}(\mathbf{R} \mathbf{R}^\top)$) while MCC only evaluates the diagonal values after column permutation. We have clarified in Section 2.4: “Unlike MCC, which only measures similarity along the main diagonal after permutation, MD also accounts for off-diagonal (dis)similarity.”
> >
> > 10. Thank you for your careful questions. We have added the results of iVAE and MISA sources color coded by age and sex in Appendix C.2: “In general, we observe that the iVAE SCVs reveal several sex clusters (e.g. SCVs $4$ and $15$) and age clusters (e.g. SCVs $5$ and $15$), but the linear fitted lines are less aligned across segments. By contrast, there is no visually obvious cluster related to sex and age from the MISA linked SCVs while the linear fitted lines are more aligned across segments. These results can be explained by the differences in the objectives of iVAE and MISA. Specifically, iVAE incorporates sex and age information during optimization, while MISA lacks access to such information. The loss function in MISA is specifically designed to align multimodal data, whereas iVAE is not explicitly designed for this purpose.”
> >
> > 11. Regarding multimodal dimension reduction, we initially used MGPCA and GICA initialization as per conventional preprocessing steps in GICA literature, but we agree that it might not be reasonable to use linear GICA while estimating nonlinear ICA sources. Thus, we only used the MGPCA initialization and removed the GICA initialization step in a new experiment. We have updated related results from the new neuroimaging experiment. Although applying linear MGPCA to reduce dimension will result in information loss, it won’t affect the identifiability of the model per se. It may, however, affect which sources are/can be identified. On the other hand, not applying dimension reduction incurs many  more parameters to be estimated (44318 input dimensions), which would be computationally expensive. It is a trade-off between information loss and computational efficiency.

---

> > > ### Author Response · Authors · 2023-11-21
> > >
> > > We thank the reviewer again for your time and suggestions. If you think we have addressed some of your concerns, we kindly ask that you consider adjusting your score accordingly. Your further feedback will be much appreciated.

---

> > > > ### Comment · Reviewer_queK · 2023-11-23
> > > >
> > > > Thank you for your response and updating the submission. The response has addressed some of my questions and concerns. I appreciate that the authors have added Appendix A that makes the proposed theoretical claims clearer, although it has not become fully clear to me how the presented proof translates to the actually implemented method (via alternating between two losses, using gradients; do the stated conditions apply).

---

### Official Review · Reviewer_dLxh · 2023-10-31

**Soundness:** 1 poor
**Presentation:** 2 fair
**Contribution:** 1 poor
**Rating:** 5
**Confidence:** 4

**Summary:**

This work discusses a nonlinear extension of independent vector analysis (Kim et al., 2006) termed DeepIVA. The authors argue, based on (Abrol et al., 2021), that DeepIVA may be useful in neuroimaging applications. Synthetic experiments, as well as experiments on neuroimaging datasets, are presented.

**Strengths:**

The authors introduce a nonlinear representation learning method which may be applicable to neuroimaging datasets, building on previous works on linear and nonlinear ICA.

**Weaknesses:**

1. The generative model assumed in the DeepIVA framework is never explicitly written in the paper.
This critical flaw not only diminishes the clarity of the presentation, but also greatly hinders the ability to assess the rigour and soundness of several claims in the paper, particularly regarding identifiability (see point below).

2. The paper lacks a proper identifiability result.
Modeling assumptions and assumptions for identifiability are never properly stated. My understanding is that the paper assumes that identifiability for DeepIVA follows trivially from previous works (e.g., iVAE).
This is, in my view, unclear, or misleading, see Questions.

(Note that, in principle, solid empirical work which does not prove identifiability results may nonetheless be relevant and worthwhile.
However, identifiability is referred to multiple times in the paper, in a way which may be deceptive if compared to other works in the nonlinear ICA literature.)

3. There is insufficient discussion of related literature on multi-view ICA, both linear and nonlinear, and its applications to neuroimaging, see Questions.

4. In Fig. 5, the reported MCC of (a maximum of roughly) 0.7 seems rather low compared to results reported in other nonlinear ICA works for identifiable models.


__________

**Edit:** After discussion with the authors, I raised my evaluation to a 5.

**Questions:**

1. The first question would be to explicitly write down the full generative model. If the work is intended to claim identifiability for this model, a Theorem proving under which assumptions said generative model is identifiable should be provided.

2. In the manuscript, it sounds like identifiability should follow trivially from previous works, such as the iVAE framework (Khemakhem et al., 2020). Even if this were the case, it would be helpful to restate and discuss the required assumptions.
Moreover, I find it hard to understand how to apply the results in (Khemakhem et al., 2020) in the context of this work. In particular, the results in (Khemakhem et al., 2020) lead to a specific kind of identification of the latent components, which is not up to permutation and element-wise nonlinear transformations: in fact, an additional linear indeterminacy remains (note also that the results refer to identifiability of sufficient statistics). It is unclear to me why the MISA network should be the right model to undo those transformations which may be unresolved in reconstructed components (with respect to the true ones), and to match components across datasets; and on what basis identifiability of the resulting two-step procedure would be guaranteed.

2. What is the relationship of the model proposed in this paper with previous literature on multi-view ICA, both linear [1, 2, 3] and nonlinear [4]?
My understanding is that multi-view ICA models the dependence of (in this paper's notation) the components of $\mathbf{s}_i$ differently---e.g., [1], eq. (1), in the linear case; or [4], eqs. (5-7) and Definition 2, in the nonlinear case.
Crucially, in the nonlinear case [4], Def. 2 introduces a technical assumption which rules out trivial cases and constitutes one of the required
assumptions for identifiability. I am confused by the lack of discussion of analogous assumptions in this work. Note also that multi-view nonlinear ICA [4] allows for identifiability in the absence of an additional auxiliary variable besides the collection of views, something the authors refer to in the Limitations section of the present paper.

4. I found the description of the neuroimaging data lacking. Could you please describe the dataset used in the experiments, and relate them to the DeepIVA model? What aspect of the neuroimaging data used in the experiments is supposed to be modelled with a statistical dependence across datasets/subjects (i.e., among components of $\mathbf{s}_i$)?
Moreover, is it task-based neuroimaging data, or resting state? If task-based, how does the proposed method compare to [1, 2] or [5]? If not (resting state), how is the statistical dependence among components of the $\mathbf{s}_i$ vector to be interpreted?

5. How would you explain the results in Fig. 5? Why is the MCC so low?

References:

[1] Richard, Hugo, et al. "Modeling shared responses in neuroimaging studies through multiview ICA." Advances in Neural Information Processing Systems 33 (2020): 19149-19162.

[2] Richard, Hugo, et al. "Shared independent component analysis for multi-subject neuroimaging." Advances in Neural Information Processing Systems 34 (2021): 29962-29971.

[3] Pandeva, Teodora, and Patrick Forré. "Multi-view independent component analysis with shared and individual sources." Uncertainty in Artificial Intelligence. PMLR, 2023.

[4] Gresele, Luigi, et al. "The Incomplete Rosetta Stone problem: Identifiability results for Multi-view Nonlinear ICA." Uncertainty in Artificial Intelligence. PMLR, 2020.

[5] Chen, Po-Hsuan Cameron, et al. "A reduced-dimension fMRI shared response model." Advances in neural information processing systems 28 (2015).

---

> ### Author Response · Authors · 2023-11-20
> **Official Comment by Authors (1/2)**
>
> We appreciate the reviewer’s insightful and constructive feedback. We include our point-to-point response as follows.
>
> 1. Thank you for the important suggestion. We have explicitly written the generative model assumed in DeepIVA, and provided a sketch of the conceptual proof showing that the generative model is identifiable up to a permutation and component-wise transformation in Appendix A. We have also updated Section 2.1 iVAE and DeepIVA parts accordingly in the revised manuscript.
>
> 2. Thank you for pointing out the multi-view ICA literature. Multi-view ICA assumes that the data can be modeled as a linear transformation of a common/shared source with an additive Gaussian noise. This assumption holds if all datasets are generated from a single shared latent variable, such as naturalistic datasets where subjects might share a similar temporal response. However, it doesn’t hold if each dataset is generated from a separate multidimensional latent variable, such as the expression level of structural and functional imaging features across subjects. The key assumption difference between multi-view ICA and DeepIVA is that sources are $\textbf{shared}$ across subjects in multi-view ICA while sources are $\textbf{linked}$ across data modalities in DeepIVA. In the context of multimodal fusion, it is more reasonable to assume that each data modality is generated by modality-specific latent variables, instead of a shared latent variable, especially for data modalities that are inherently heterogeneous (such as imaging and genomics). Thus, we believe that DeepIVA is more capable in multimodal fusion tasks, as it aims to identify a set of linked sources for each modality.
>
> We have added a discussion to distinguish our approach and multi-view ICA in Section 1 Introduction: “Recent studies on multi-view BSS assume that observations from different views originate from a shared source variable and distinct additive noise variables (Richard et al., 2020; 2021; Pandeva & Forre, 2023; Gresele et al., 2020). However, in the context of multimodal fusion, it is more reasonable to assume that each modality is generated by modality-specific latent variables which, in turn, are linked across modalities, rather than a shared set, especially for data modalities that are inherently heterogeneous.”
>
> 3. Sorry for the lack of details about the neuroimaging dataset. We use structural MRI data (gray matter segmentation maps from T1-weighted images) and the voxel-wise amplitude of low frequency fluctuations (ALFF) from a resting-state fMRI dataset collected on the same subjects. The statistical dependence between modalities is modeled using subject expression levels. While we assume expression levels change (i.e., are nonstationary) according to age and sex groups, we assume the statistical dependence between data modalities remains the same and, thus, can be interpreted as structural and functional relationships associated with each latent source.
>
> We have revised Section 2.3 Neuroimaging Data: “We utilize the UK Biobank dataset (Miller et al., 2016) $\mathbf{X}\in \mathbb{R}^{N \times V \times M}$ including two imaging modalities T1-weighted sMRI and resting-state fMRI ($M=2$) from $2907$ subjects ($N=2907$). We preprocess sMRI and fMRI to obtain the gray matter tissue probability segmentation (GM) and amplitude of low frequency fluctuations (ALFF) feature maps, respectively. Each GM or ALFF feature map includes $44318$ voxels ($V=44318$). Here, we use age and sex groups as auxiliary information, assuming that sources within each modality are conditionally independent given the age and sex group. This assumption is based on studies demonstrating the significant impact of age and sex on both brain structure and function (Raz et al., 2004; Good et al., 2001; Ruigrok et al., 2014).”

---

> ### Author Response · Authors · 2023-11-20
> **Official Comment by Authors (2/2)**
>
> 4. Regarding the MCC results, please first note that the per-modality per-segment MCC metric evaluates the MCC in the same way as in the nonlinear ICA literature, and these per-modality per-segment MCC values for 5 sources (>0.9) are very close to those reported in Khemakhem et al. 2020. Furthermore, in order to account for cross-modal linkage and cross-segment consistency, we propose a new approach to measure the MCC values (see Section 2.4 for details), and the proposed per-modality MCC, per-segment MCC and aggregated MCC metrics may be lower because the iVAE fails to align linked sources across modalities or consistent sources across segments.
>
> MCC values are lower when we estimate more latent sources (15 sources) with the same sample size (2800 samples). Our hypothesis is that the sample size was not sufficient to accurately recover more latent sources. We repeated the experiments with a double sample size (5600 samples) and an intermediate problem scale (10 sources). According to the new result, we observe that 1) model performance decreases as the number of sources increases when the sample size stays the same; 2) DeepIVA performance improves as the sample size increases when the number of sources stays the same.
>
> Please see Figure 4 and Section 3.1 in the revised manuscript: “We perform a systematic evaluation of model performance across different data-generating configurations by varying both the problem scale ($5$, $10$ and $15$ sources) and the sample size ($2800$ and $5600$ samples). The aggregated MD and MCC metrics are shown in Figure 4. Remarkably, DeepIVA outperforms iVAE and MISA in every configuration, showcasing its superior performance across all evaluated scenarios. Within each panel, we observe a consistent drop in model performance as the number of latent sources increases, suggesting that the optimization problem becomes more challenging as the latent dimension increases. Across horizontal panels, the DeepIVA performance improves for configurations with $10$ and $15$ sources when the sample size increases from $2800$ to $5600$, indicating that a larger sample size is necessary to better recover sources in a harder problem.”

---

> > ### Author Response · Authors · 2023-11-21
> >
> > We thank the reviewer again for your time and suggestions. If you think we have addressed some of your concerns, we kindly ask that you consider adjusting your score accordingly. Your further feedback will be much appreciated.

---

> ### Comment · Reviewer_dLxh · 2023-11-22
> **How to align linked sources?**
>
> I thank the authors for their reply, and for adding Appendix A.
>
> While it helps clarify the method, the identifiability theory provided therein (mostly relying on results in (Khemakhem et al., 2020)) still leaves me with a few doubts. I will detail this below. Please let me know in case I am misunderstanding something.
>
> **Distinction between shared and linked.** May I kindly ask you to elaborate on the distinction between shared and linked sources? Does "linked" mean that there is some kind of statistical dependence? In multi-view ICA, each view is generated by mixing a corrupted version of some shared sources ($\mathbf{s}$ are the shared sources, $\mathbf{s}+\mathbf{n}_i$ would be the corrupted version for the $i$-th view). So the $i$-th component of the corrupted sources is not independent across different views. Can I think of these corrupted components, or sources, as linked according to your terminology? My impression is that these could be considered linked, albeit possibly with a different kind of statistical dependence from the one in your equation (3).
>
> **MISA in the nonlinear case.** Based on my understanding, the MISA procedure relies on "all-order statistics" to match linked sources. However, in nonlinear ICA (including the iVAE results), an identifiable model may separate the true sources, but element-wise non-linearly distort them w.r.t. the true ones. This element-wise distortion has, in principle, the capability to transfrom any univariate distribution into a completely different one, thereby potentially modifying statistics of all orders, and (in my understanding) destroy the signal MISA relies on to operate the matching. (Note that this problem does not arise for linear models, for which MISA was originally developed.) Separately training different models on different datasets may recover sources which are distorted up to distinct element-wise distortions. Based on this, I'm having difficulty understanding the rationale behind the possibility of matching linked sources.
>
> **fMRI.** Thanks for the provided explanation. May I kindly ask you to further clarify what the linked sources are expected to capture, i.e., what is the meaning of linked sources across data modalities? Should I think of them as capturing some subject-specific characteristics influencing both functional activity in fMRI and structural properties captured by sMRI? Can I think of a generating process in which a first variable captures properties which depend on subject identity, $I_i$, and in turn this influences both structural properties $S_i$ and functional activity $F_i$, thus rendering them dependent? I would represent this as a DAG with arrows $S_i \leftarrow I_i \rightarrow F_i$. Is this a correct picture? If so, I believe that this would strengthen the connection to multi-view ICA (see e.g. Fig. 2 in (Gresele et al., 2019)), related to my concerns expressed above.

---

> > ### Author Response · Authors · 2023-11-23
> >
> > We thank the reviewer for the thoughtful questions and clarify these follow-up questions as below.
> >
> > **Distinction between shared and linked**:
> >
> > We consider sources following a multivariate distribution with second- or all-order statistical dependence to be “linked”.
> >
> > Shared sources, on the other hand, are sources that share an identical underlying variable. For example, let two latent variables $s_1$ and $s_2$ share the same underlying variable $s$, $s_1=s+n_1$, $s_2=s+n_2$. Altogether, $s$, $n_1$ and $n_2$ make up a 3D distribution of 3 independent variables. Projecting from 3D to 2D (a transformation of variables) causes the resulting $s_1$ and $s_2$ to become linearly dependent, since they share the same underlying $s$, and $n_1$ and $n_2$ are independent, meaning $p(s_1,s_2)$ contains only second order statistics. Thus, complicated dependence relationships that transcend second-order statistics cannot be modeled accurately with shared sources. When linear relationships account for all of the dependence in a dataset then shared sources may furnish a good approximation to linked sources.
> >
> > Intuitively, if two sets of observations $x_1$ and $x_2$ are generated by two different variables $s_1$ and $s_2$, using a shared source with additive noise ($s+n_1$ and $s+n_2$) may not be sufficient to model $s_1$ and $s_2$. Thus, it is more flexible and appropriate to model each modality or dataset with a separate random variable.
> >
> > **MISA in the nonlinear case:**
> >
> > Point-wise nonlinearities do not alter the dependence between variables. The theory of copulas covers that concept extensively (see Sklar’s theorem): any continuous joint PDF can be represented as $p(s_1)p(s_2)c(F(s_1), G(s_2))$, where $c(\cdot, \cdot)$ is the copula density and $F$ and $G$ are the marginal CDFs of $s_1$ and $s_2$, respectively. The copula density $c(\cdot, \cdot)$ captures all of the dependence between $s_1$ and $s_2$, while $p(s_1)$ and $p(s_2)$ can be nonlinearly altered arbitrarily ($F$ and $G$ will compensate the nonlinear transformations automatically). The conclusion is that the point-wise nonlinearities from iVAE do not alter the dependence between linked sources.
> >
> > Thus, one could intentionally transform all final iVAE sources into Kotz marginals and then pursue combinatorial optimization in MISA to detect the links. Note, however, that Kotz marginals don’t have a well-defined closed-form, which further motivates the interweaved optimization of the MISA loss (MLE) to gradually induce such marginals on the iVAE latents.
> >
> > **fMRI:**
> >
> > Linked sources are expected to capture statistically dependent expression levels of structural and functional features over subjects. Note that sMRI and fMRI data are separately collected and characterize different aspects of the brain. The assumption of shared variables may not be sufficient to characterize these two heterogenous imaging properties. In a general sense, it is unlikely that a single continuous number (the random variable $I_i$) suffices to capture all combinations of co-occurring structural and functional expressions across individuals nor all possible “identity” groupings of individuals (e.g., the XOR problem in classification).
> >
> > Here, we are interested in studying different heterogeneous data modalities (structural and functional imaging data) assuming a generative model with multimodal conditionally independent joint latent variables and modality-specific nonlinear mixing processes. Borrowing the reviewer’s notation, this can be represented as: $I_i^S \rightarrow s_1$, $I_i^F \rightarrow s_2$, effectively modeling the subject-specific “intensity” of the brain pattern from each modality with a different random variable. If it were the case that $p(I_i^S, I_i^F)$ is fully explained by linear dependence, then a model of shared sources might be useful, but that is currently unknown, and it is necessary to develop approaches that capture higher-order dependence.

---

> ### Comment · Reviewer_dLxh · 2023-11-23
> **Thanks for your reply; raising my evaluation**
>
> Thank you for your reply, which answered some of the points I raised.
>
> **Distinction between shared and linked; fMRI.** My understanding of your reply is that corrupted sources with additive noise, $\mathbf{z}_1 := \mathbf{s}+\mathbf{n}_1$ and $\mathbf{z}_1 := \mathbf{s}_2+\mathbf{n}_2$, can only model simple (linear) dependence between $\mathbf{z}_1, \mathbf{z}_2$. More complicated statistical dependence cannot be modelled accurately by the additive model in (Richard et al., 2020), whereas the model you use may provide more expressivity. This sounds like a valid argument, where it appears that there is a strong connection between corrupted sources $\mathbf{z}_1, \mathbf{z}_2$, as defined above, and the linked sources you consider; your considered case may be more expressive and, potentially, more useful for neuroimaging data:
>
> > If it were the case that $p(I^S_i, I^F_i)$ is fully explained by linear dependence, then a model of shared sources might be useful, but that is currently unknown, and it is necessary to develop approaches that capture higher-order dependence.
>
> I accept the explanation in principle, although it would be interesting to have some empirical check and comparison of the two methods, to verify whether approaches that capture higher-order dependence (and nonlinear mixing), as yours, allow for an improved performance or discovery of interesting biomarkers.
>
> I would suggest discussing this in more detail in the paper, particularly because more complex (non-additive) corruptions $\mathbf{z}_i := \mathbf{g}(\mathbf{s},\mathbf{n}_i)$ have already been considered in the literature on the theory of nonlinear multi-view ICA (Gresele et al., 2019).
>
> **MISA in the nonlinear case.** Thank you for the provided explanation. I agree that element-wise transformations cannot erase dependence among the linked sources. I would appreciate it if you could provide a self-contained proof along with a formal illustration demonstrating how this argument leads to identifiability for the method you proposed (the current presentation of the proof in Appendix A may be seen as not fully explicating all the formal passages, as also observed by other reviewers).
>
> I appreciate the authors' efforts in replying to my questions. Since the answers clarified some of the points I raised, I will raise my evaluation.

---

### Official Review · Reviewer_Uwfa · 2023-11-01

**Soundness:** 2 fair
**Presentation:** 2 fair
**Contribution:** 2 fair
**Rating:** 3
**Confidence:** 4

**Summary:**

The manuscript introduces a method that combines iVAE and MISA in order to learn linked and identifiable latent sources from multiple data modalities. Comprehensive experiments on both synthetic and real-world datasets show that the proposed method outperforms both iVAE and MISA. However, the contribution is limited due to its lack of novelty beyond the combination.

**Strengths:**

1. The experiments are comprehensive.

**Weaknesses:**

1. The contribution is limited. It is essentially a combination of two existing methods (MISA and iVAE) without any theoretical results. Since the key contributions of MISA and iVAE are their identifiability proofs instead of the estimation (e.g., the estimation model of iVAE is basically the vanilla VAE), combining their estimation methods without any theoretical contribution might be incremental.

2. Almost half of the manuscript is dedicated to the introduction of existing works, and the other half focuses on experiments. Meanwhile, only **half a page** describes the proposed method. The brevity of the description might be understandable if the contribution is simply a combination of existing methods. However, if that's not only a combination, a more detailed explanation regarding the motivation and unique contributions of the proposed method is necessary.

**Questions:**

In the section of the conclusion, it is mentioned that there exist some recent works on nonlinear ICA without auxiliary variables. However, in the introduction, only methods with auxiliary variables are introduced. Any specific reason for that?

---

> ### Author Response · Authors · 2023-11-20
>
> $\textbf{Q1}:$ More detailed explanation regarding the motivation and unique contributions.
>
> $\textbf{A1}:$ We thank the reviewer for requesting clarification on the motivation and contributions of our proposed method.
>
> Regarding motivation, we have revised the manuscript to explain why we are interested in multimodal fusion in Section 1 Introduction: “Jointly analyzing two imaging modalities can uncover cross-modal relationships that cannot be detected by a single imaging modality, providing new insights into structural and functional interactions in the brain and its disorders (Calhoun & Sui, 2016).”
>
> Also, another reason we choose MISA is that we want to utilize its flexible subspace modeling to further solve nonlinear independent subspace analysis (ISA) problems. We have added one more reason why we choose MISA to align latent sources in Section 4 Discussion: “We plan to extend our proposed method from nonlinear IVA problems to nonlinear ISA problems, aiming to capture source dependence by leveraging higher-dimensional subspaces.”
>
> Regarding contributions, our study is an initial step towards learning linked and identifiable sources from multimodal data. We empirically demonstrate that DeepIVA can learn linked and identifiable sources by unifying iVAE and MISA, and “the idea of combining iVAE and MISA is new”, as acknowledged by reviewer queK. We would like to point out that our proposed method is not “simply a combination of existing methods”. We modified the iVAE encoder architecture to process data features and auxiliary information separately, as noticed by reviewer dLxh. Additionally, the linear demixing matrix in the original MISA framework was replaced by a nonlinear encoder, and we adjusted the loss term accordingly as described in Section 2.1 Deep Independent Vector Analysis.
>
> We have updated Figure 1 to illustrate the extra details of our method, including the data input and the encoder layer. We have also updated Section 2.1 Deep Independent Vector Analysis to explain the architecture change: “Additionally, since MISA is not designed to handle auxiliary information, we modify the original encoder architecture to distinguish between data features $\mathbf{x}^{m}$ and auxiliary variables $\mathbf{u}$ such that 1) the iVAE updates model parameters with respect to both $\mathbf{x}^{m}$ and $\mathbf{u}$ at the input layer, and 2) the MISA updates only those pertaining to $\mathbf{x}^{m}$ but not $\mathbf{u}$. The original iVAE model uses a single input layer taking the concatenated $\mathbf{x}^{m}$ and $\mathbf{u}$. In DeepIVA, we split this layer into two: one for data features $\mathbf{x}^{m}$ and another for auxiliary variables $\mathbf{u}$. The parameters with respect to $\mathbf{u}$ will only be updated at the iVAE training step but will remain frozen at the MISA training step. Also, the inputs for the auxiliary variables are set to $0$ during MISA training to ensure no influence from the frozen weights.”
>
> Furthermore, we have restated the identifiability theory in Khemakhem et al., 2020, and provided a conceptual sketch of proof showing that the generative model assumed in DeepIVA is identifiable up to a permutation and component-wise transformation in Appendix A. We have also updated Section 2.1 iVAE and DeepIVA parts accordingly in the revised manuscript.
>
> $\textbf{Q2}:$ Related works on nonlinear ICA without auxiliary variables.
>
> $\textbf{A2}:$ We introduce methods relevant to the current work in the introduction section and mention methods worth exploring in the future work section. We are interested in the application of neuroimaging analysis, and the iVAE has been shown to work on an fMRI dataset (Khemakhem et al. 2020) among the mentioned papers. Thus, we chose iVAE for this present work, and we will keep exploring other methods.

---

> > ### Author Response · Authors · 2023-11-21
> >
> > We thank the reviewer again for your time and suggestions. If you think we have addressed some of your concerns, we kindly ask that you consider adjusting your score accordingly. Your further feedback will be much appreciated.

---

> > > ### Comment · Reviewer_Uwfa · 2023-11-22
> > >
> > > Thanks so much for your response. I'm still not fully convinced regarding the significance of the theoretical contribution compared to previous works, since the identifiability of the proposed method (iVAE+MISA) is mostly based on existing results. Thus, I would like to maintain my score for now.

---

### Meta-Review · Area_Chair_B2Ue · 2023-12-02

**Metareview:**

**Summary:** The manuscript presents a method that combines iVAE and MISA to learn identifiable latent sources across multiple data modalities. Experimental results demonstrate superior performance compared to selected individual baselines.  However, the paper lacks a strong theoretical justification and clear proof for this combination.  Implementation details are unclear.  The proposed method's contribution is not detailed, leading to a lack of clarity and arousing doubts about its novelty, as it appears to be a basic combination of existing methods without substantial theoretical advancements.

During the rebuttal, the reviewers discussed that the authors discussion helped them understand the work better, but still had concerns about the main theoretical claims of the paper.  Thus, I recommend the rejection of the paper.


**Strengths:** The manuscript presents a novel nonlinear representation learning method that shows potential for neuroimaging datasets.  The combination of iVAE and MISA is a new approach, and aligning the latent representation using MISA is an interesting alternative to self-supervised learning methods. However, the exploration of MISA in multi-modal settings is lacking. Numerical experiments on synthetic and neuroimaging data demonstrate improved inferred representations compared to iVAE and MISA, both in cross-modal and unimodal settings. The paper is well-written, with clear descriptions of the proposed method, metric formulations, and experimental setup.

**Weaknesses:**  The weaknesses identified in the paper include the lack of a theoretically grounded justification and clear proof for the combined use of MISA and iVAE in a cross-modal setting, as well as a missing definition for 'identifiability'.  The implementation details are unclear, relying on a restrictive assumption of data independence within each modality, and the newly introduced MC measure lacks adequate justification over the MCC measure. The paper also fails to explicitly mention the generative model used in the DeepIVA framework, does not clarify the identifiability result or necessary conditions, and lacks sufficient exploration of related literature on multi-view ICA. Additionally, the proposed method is not well-explained, causing a lack of clarity and understanding of its contribution, and the reviewers question its novelty, perceiving it as a basic combination of existing methods without a substantial theoretical contribution. The paper's excessive focus on existing works further contributes to the lack of clarity and comprehensive understanding of the proposed method.

**Justification For Why Not Higher Score:**

The main reason for rejection is the lack of theoretical justification, and in the current setup, the proposal appears to be only incrementally beneficial. Despite the appendix addressing some concerns, the reviewers still believe that additional work is necessary to ensure the theoretical justification is suitable for publication.

**Justification For Why Not Lower Score:**

N/A

---

### Decision · Program_Chairs · 2024-01-16

Reject